# Inhibitor-based modulation of huntingtin aggregation mechanisms mitigates fibril-induced cellular stress

Greeshma Jain [1,5], Marina Trombetta-Lima [2,3,5], Irina Matlahov[1], Hennrique Taborda Ribas[2,4], Tingting Chen[2], Raffaella Parlato[1], Giuseppe Portale [1], Amalia M. Dolga [2] ✉ & Patrick C. A. van der Wel [1] ✉

Huntington's disease (HD) is a neurodegenerative disorder in which mutated fragments of the huntingtin protein (Htt) undergo misfolding and aggregation. Since aggregated proteins can cause cellular stress and cytotoxicity, there is an interest in the development of small molecule aggregation inhibitors as potential modulators of HD pathogenesis. Here, we study how a polyphenol modulates the aggregation mechanism of huntingtin exon 1 (HttEx1) even at sub-stoichiometric ratios. Sub-stoichiometric amounts of curcumin impacted the primary and/or secondary nucleation events, extending the pre-aggregation lag phase. Remarkably, the disrupted aggregation process changed both the aggregate structure and its cell metabolic properties. When administered to neuronal cells, the 'break-through' protein aggregates induced significantly reduced cellular stress compared to aggregates formed in absence of inhibitors. Structural analysis by electron microscopy, small angle X-ray scattering (SAXS), and solid-state NMR spectroscopy identified changes in the fibril structures, probing the flanking domains in the fuzzy coat and the fibril core. We propose that changes in the latter relate to the presence or absence of polyglutamine (polyQ) β-hairpin structures. Our findings highlight multifaceted consequences of small molecule inhibitors that modulate the protein misfolding landscape, with potential implications for treatment strategies in HD and other amyloid disorders.

Proper protein folding is vital for living organisms, with small errors resulting in misfolded protein structures that can lead to lethal outcomes in various human diseases[1,2]. In the central nervous system, pathogenic protein deposition is associated with a range of neurodegenerative disorders[1]. One such neurodegenerative disease is Huntington's disease (HD) which is one of a family of CAG repeat expansion diseases, in which mutations cause the expansion of a polyglutamine (polyQ) stretch in a specific protein[3–5]. In HD, the polyQ stretch is located near the N-terminus of the mutant huntingtin protein, within its first exon (HttEx1); see Fig. 1a. Disease risk is predicted by the extent of polyQ expansion beyond a disease-specific threshold, with the polyQ length inversely correlating with age-of-onset. N-terminal

[1]Zernike Institute for Advanced Materials, Faculty of Science and Engineering, University of Groningen, Groningen, The Netherlands. [2]Department of Molecular Pharmacology, Faculty of Science and Engineering, Groningen Research Institute of Pharmacy (GRIP), University of Groningen, Groningen, The Netherlands. [3]Department of Biomedical Sciences of Cells and Systems, Section Molecular Cell Biology, University of Groningen, University Medical Center Groningen, Groningen, The Netherlands. [4]Graduate Program in Biochemistry Sciences, Department of Biochemistry and Molecular Biology, Federal University of Paraná, Curitiba, PR, Brazil. [5]These authors contributed equally: Greeshma Jain, Marina Trombetta-Lima. ✉e-mail: a.m.dolga@rug.nl; p.c.a.van.der.wel@rug.nl

**Fig. 1 | Structures of Htt exon 1, curcumin and Thioflavin T (ThT). a** Domain structure of HttEx1, showing the central polyQ segment (blue) as well as the N-terminal (htt[NT]; orange) and C-terminal flanking segments (proline-rich domain; PRD; green). **b** Model of the previously determined fibril architecture of Q44-HttEx1 forming single-filament fibrils. Indicated domains: htt[NT] (orange; α-helix), polyQ (red and blue; β-sheet), and PRD (green; PPII helices). **c** Chemical structures of curcumin and (**d**) Thioflavin T. Panel b was adapted from ref. 19 under its CC open-access license.

fragments of Htt with an expanded polyQ segment are found as intraneuronal deposits in patients[4,6]. Such protein aggregates are commonly seen as hallmarks of HD pathology[4,5] and potential biomarkers, driving the development of aggregate-binding ligands for PET imaging[7] and methods that detect their seeding ability in cerebral spinal fluid (CSF)[8,9]. The exact role of the misfolded and aggregated proteins in HD pathogenesis remains a topic of debate. Htt-derived protein aggregates have been implicated in a variety of toxic mechanisms and disruptions of vital metabolic pathways[5,10]. Yet, there is also evidence that aggregated Htt fragments can form aggregates that seem less pathogenic[11–13]. These variations in pathogenic properties have led to much discussion of whether aggregates or inclusion bodies are toxic or a rescue mechanism. Unification of the diversity of findings may be found in the recognition that these proteins can form a range of different types of aggregates, which likely differ in their interactions and pathogenic properties[11,12,14,15]. For instance, it may be that large inclusions detectable by (diffraction-limited) microscopy have limited cytotoxic consequences, while smaller fibrillar or oligomeric species (detected in cryo-EM or super-resolution microscopy[16]) have a more pronounced pathogenic footprint.

There is considerable interest in understanding as well as modulating HD-related protein misfolding and aggregation, both for diagnosis and treatment[7,17,18]. These efforts benefit from recent breakthroughs in the use of structural techniques to probe the protein aggregates. In prior structural studies it has been found that the misfolded aggregates formed by mutant HttEx1 are fibrillar, with a dense and rigid β-sheet core formed by the expanded polyQ segment[15,19–23]. The architecture of this polyQ core displays all the hallmarks of other amyloid-like protein aggregates, such as those from other neurodegenerative diseases[2,24]. Previously, we proposed a structural model for mutant HttEx1 fibrils (Fig. 1b), in which the polyQ forms a β-sheet rigid core, with N- and C-terminal 'flanking' domains exposed on the fibril surface as a 'fuzzy coat'[14,19,23]. This model is supported by diverse techniques, including solid-state NMR (ssNMR) and cryo-EM[15,20,25–30]. The fibril structure has several notable features of relevance to the current study, given expected implications for the fibrils' biological properties. First, the polyQ segment itself forms a rigid dehydrated core structure, within which the Q44-HttEx1 variant was found by ssNMR to feature a β-hairpin fold[27]. This translated into a 6-nm fibril

width as observed via negative-stain transmission EM (TEM). As previously discussed[14,19,27], we take this TEM dimension to reflect the width of the impenetrable polyQ core, excluding the flexible non-β-sheet flanking segments that permit penetration of water and stain molecules[14,25,31]. These flanking segments are the N-terminal N17 or Htt[NT] segment, and the C-terminal proline-rich domain (PRD; Fig. 1a).

A variety of intervention strategies can be envisioned to slow the progression of diseases like Alzheimer's, Parkinson's, and HD: e.g. antibody-based therapeutics, protein-lowering strategies, and small molecule aggregation modulators[17,32–34]. Much recent effort has gone into HD protein-lowering approaches, but we hypothesize that methods that target the downstream aggregation process may be a valuable complement, also given recent challenges in clinical tests of Htt lowering in patients[35]. Many small molecules have been studied as modulators of Aβ, α-synuclein, and HttEx1 aggregation[36–38]. A common approach is based on naturally occurring polyphenols[39], which also have anti-oxidative and anti-inflammatory properties, and show therapeutic potential in neurodegenerative disorders[40,41]. Curcumin (Fig. 1c) belongs to the curcuminoid family and is derived from the *Curcuma longa* plant. It has been used medicinally for many years because of its antioxidant, anticancer, anti-inflammatory, and neuroprotective activity, although its limited bioavailability can present a challenge in clinical applications[42,43]. It has yielded promising results in studies on Alzheimer's disease and α-synuclein aggregation[39,44–47].

Here we study the modulation of HttEx1 misfolding and aggregation by curcumin and probe different aspects of how it can modulate amyloid toxicity. In recent years there has been a growing interest in the mechanistic underpinnings of how inhibitors can act on different stages of the aggregation process[48,49]. Part of this interest derives from the observation that different misfolded species along the aggregation pathway, as well as distinct fibril polymorphs, can have widely varying toxic properties[50,51]. Thus, we were interested in the details of how an aggregation modulating compound changes the aggregation process and in particular the fibril polymorphism of HttEx1. We observed that curcumin can delay pathogenic fibril formation, in aggregation studies in vitro. The fibrils formed in the presence of an inhibitor displayed distinct structural and functional features. By EM, SAXS, and ssNMR measurements, we identify and compare fibril polymorphs formed in presence and absence of inhibitor. Structurally, a key difference originates from a different fibril core dimension, presumably dictated by a changed supramolecular arrangement of the polyQ segment. Crucially, the aggregates also differed in their effects on the metabolism and viability of cultured neuronal cells, despite being taken up into the cells, indicating the formation of a polymorph with reduced abilities to induce cellular stress as a consequence of the inhibitory action.

## Results

### Curcumin perturbs the aggregation kinetics of HttEx1

To study the kinetics of HttEx1 aggregation in presence and absence of curcumin we performed an amyloid-sensitive dye-binding assay using the Thioflavin T (ThT) fluorophore. ThT (Fig. 1d) is commonly used to detect and quantify β-sheet-rich amyloid fibrils, based on its ability to selectively bind to such fibrils, including polyQ protein aggregates. Monomeric HttEx1 containing a 32- or 44-residue polyQ domain (Q32-HttEx1 or Q44-HttEx1) was obtained by proteolytic cleavage of a soluble fusion protein[19,26]. From previous studies[11,12,14,19], we know that HttEx1 aggregation can produce different polymorphs, with notable effects of the concentration and temperature[19]. Here, we employed a concentration (61 μM) in PBS (pH 7.4) and (room) temperature where we reproducibly can form a dominant type of polymorph[19]. In absence of curcumin, fibril formation was detected as an increase in ThT fluorescence, with an emission maximum at 490 nm (Fig. 2a), after a lag-phase of 5-6 h. In presence of curcumin (Fig. 2a) we observed an increased lag-phase, with the increase extending with increasing curcumin concentrations. The extended lag phase suggests that curcumin

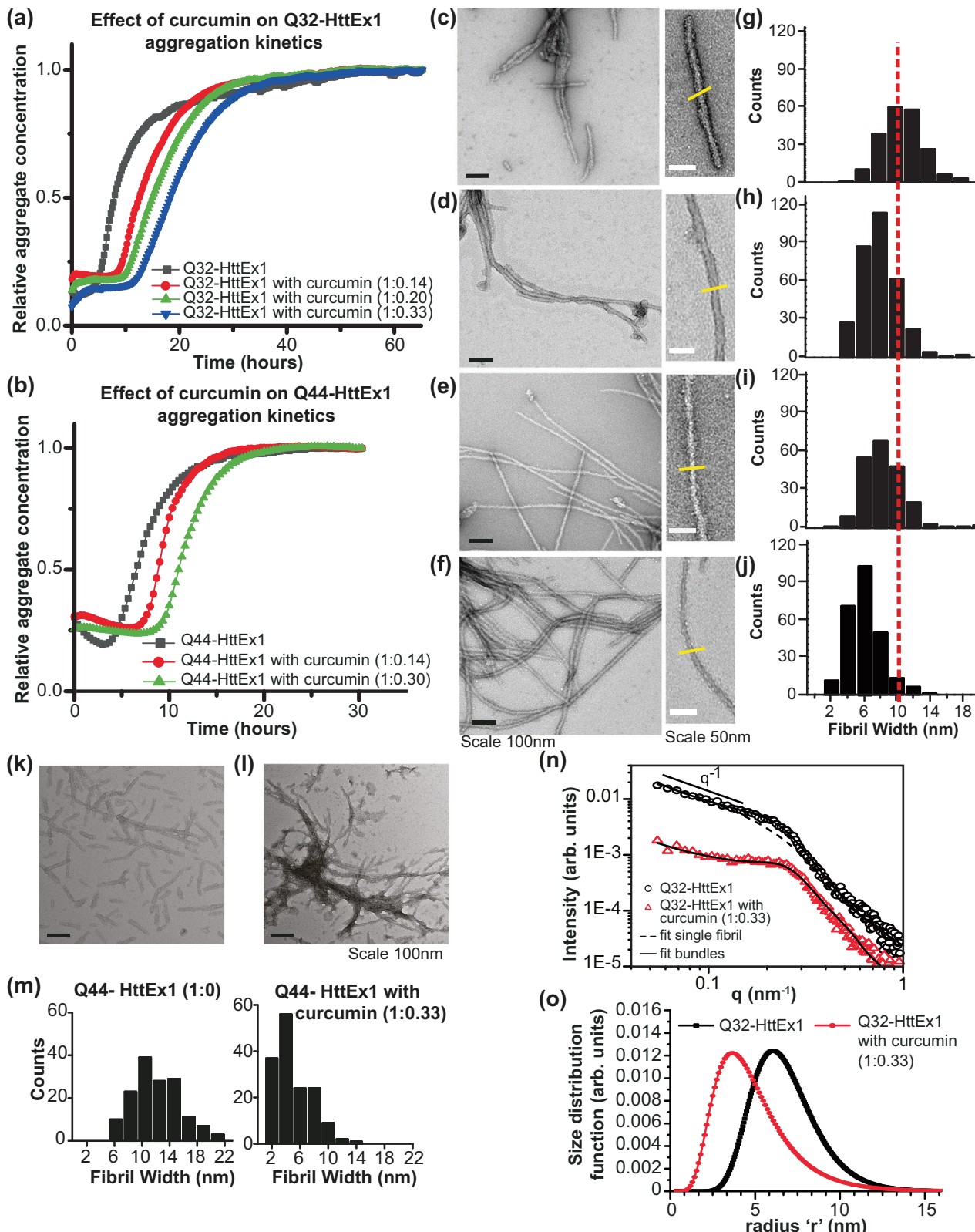

could be perturbing nucleation processes early in the misfolding and aggregation process[52]. Prior work has shown prominent roles for both primary and secondary nucleation in HttEx1 aggregation[19,38], which is also supported by an analysis of the current (uninhibited) Q32-HttEx1 aggregation kinetics (Supplementary Fig. 1a, b). The curcumin fluorescence curves also show a different steepness in the fibril growth phase. Analogous mechanistic changes due to sub-stoichiometric

amounts of curcumin were observed also for HttEx1 containing a 44-residue polyQ domain (Q44-HttEx1; Fig. 2b). It is important to note that a very precise comparison between the fluorescence curves with and without curcumin is complicated by the fact that curcumin itself shows a dramatic increase in fluorescence upon fibril formation, as it also binds the fibrils (Supplementary Fig. 1c–g)[53]. The fluorescent properties of ThT and curcumin are difficult to disentangle, but since both

**Fig. 2 | Fluorescence assays and structural analysis of HttEx1 aggregation in presence and absence of curcumin. a** Q32-HttEx1 (61 µM) aggregation was monitored by ThT assays, measuring fluorescence in the presence and absence of curcumin (molar ratio indicated). **b** Q44-HttEx1 (67.5 µM) aggregation by ThT fluorescence in the presence and absence of curcumin. The curves indicate the mean of the triplicate values and the figure with the error bars is plotted in Supplementary Fig. 2a, b. **c–f** Negative stain TEM micrographs of 61 µm Q32-HttEx1 fibrils, formed **c** without curcumin; **d** with 8.5 µM curcumin; **e** with 12 µM curcumin; **f** with 20 µM curcumin. Black scale bars indicate 100 nm, and white bars show 50 nm. Micrographs shown in **c**, **d**, **e** and **f** are representative data from a total of 58, 41, 32 and 36 images obtained, respectively. **g–j** TEM-based fibril core width histograms corresponding to samples in **c–f**. The vertical dashed line marks the average width of fibrils formed in absence of curcumin. The yellow lines perpendicular to the fibril axis mark locations of fibril width measurements. **k,l** TEM micrographs of 50 µM Q44-HttEx1 fibrils **k** formed in absence and **l** in presence of 16.6 µM of curcumin at room temperature. Micrographs shown in **k** and **l** are representative data from a total of 28 and 45 images obtained, respectively. **m** Fibril width analysis of Q44-HttEx1 fibrils prepared without curcumin and in presence of curcumin. **n** SAXS profile and **o** size distribution profile for Q32-HttEx1 fibrils prepared in presence and absence of curcumin. Dashed and solid black lines show fits of the data with models of either isolated or bundled fibrils (see text).

show an analogous gain in fluorescence on fibril formation, we used them here interchangeably (see also Supplementary Fig. 1d).

## Curcumin inhibition results in a different fibril morphology
The aggregation kinetics assay demonstrated that curcumin interacts with the HttEx1 protein, modulating the nucleation processes and causing a delay in aggregation. Still, after the extended lag phase, amyloid fibrils appear to be detected by the increased fluorescence. To verify that the increase in fluorescence intensity stems from amyloid fibrils, and probe for possible differences in fibril morphology, we employed negative stain TEM. Figure 2c–f, k, I show TEM micrographs obtained for Q32-HttEx1 and Q44-HttEx1 aggregated in absence and presence of curcumin. The Q32-HttEx1 fibrils formed in absence of curcumin (Fig. 2c) had a fibrillar appearance with a typical width of 10 nm (Fig. 2g). In line with prior studies[12,14,19,38], HttEx1 fibrils had a tendency to cluster or bundle together (Supplementary Fig. 2d). Notably, when formed in presence of increasing amounts of curcumin, the fibrils show changes in both width of individual filaments and their propensity to bundle together (Fig. 2d–f, h–j). Compared to the fibrils formed in absence of curcumin, the fibrils formed in presence of curcumin had a more bundled appearance (Supplementary Fig. 2e). Fibrils formed in presence of curcumin are relatively narrow (·4 nm) (Fig. 2h–j), with the observed width distribution dependent on the curcumin concentration, relative to fibrils formed in absence of curcumin (Fig. 2g). Also Q44-HttEx1 aggregated in presence curcumin (Fig. 2k–m) yielded narrower fibrils, compared to those formed in absence of curcumin. Thus, TEM validates the presence of fibrillar aggregates, but also revealed a clear curcumin-dependent change in fibril width, morphology, and supramolecular interactions.

To further assess changes in supramolecular fibril structure and dimensions small angle X-ray scattering (SAXS) was performed. Figure 2n shows the background corrected experimental SAXS profiles for Q32-HttEx1 fibrils formed with and without curcumin.

Generally, three different regions can be detected: the low $q$ region ($q < 0.2$ nm$^{-1}$); the intermediate $q$ region (0.12 nm$^{-1}$ $< q <$ 0.3 nm$^{-1}$); and the high $q$ region ($q > 0.3$ nm$^{-1}$), the so-called Porod-region. In the low q region, the SAXS curves showed a power law decay of q$^{-\alpha}$. In the case of linear rod-like objects, an exponent $\alpha = 1$ is observed, and the intensity follows the power law decay in the range $1/R < q < 2\pi/L$, where $R$ and $L$ are the radius and length of the rods. As shown in Fig. 2n, the observed power law decay of the SAXS curves is consistent with the presence of linear fibrillar structures for length scales between $\approx 1/q^* = 1/$ (0.12 nm$^{-1}$) $\approx 8$ nm and $\approx 2\pi/q_{min} = 1/(0.05$ nm$^{-1}) \approx 125$ nm, where $q_{min}$ is the minimum accessible $q$-value and $q^*$ is the scattering vector value where the intensity slope in the log-log plot changes from the $q^{-1}$ trend. When comparing the experimental intensity with the simulated profile for a system composed by diluted, non-interacting fibrils[54] dashed line in Fig. 2n, it is clear that the intensity profiles show the presence of a broad peak in the intermediate q-region and located at $q_{max}$ - 0.25 nm$^{-1}$. Such a correlation peak is due to the existence of an average inter-fibrillar distance ($d_{f-f} = 2\pi/q_{max}$) as a consequence of the lateral aggregation (i.e., bundle formation) of the fibrils. The broad nature of this correlation peak implies that the correlation between fibrils is only short-ranged,

and could either be coming from bundles formed by very few fibrils or from bundles formed by many fibrils but including a large amount of defects such as kinks or breaks along the fibril length. The peak appears more pronounced for the sample prepared in the presence of curcumin, indicating that the bundling behavior is more pronounced in this case. In the high $q$ region, after the correlation peak and for length scales of the order of the fibrils' width and smaller, the SAXS intensity is dominated by the scattering contribution from the fibrils cross-section, and the intensity decay changes from −1 to −3.2. The cross-sectional radius distribution and the lateral packing of the fibrils without and with curcumin can be accurately calculated by fitting the experimental intensity using the scattering function for long rod-like objects (see SI for details). In agreement with TEM observations, the average size of the Q32-HttEx1 fibrils appears reduced in the presence of curcumin (Fig. 2o). The average size (fibrillar cross-section estimated at 2*$r$, where $r$ is the fibril radius) for the Q32-HttEx1 is 12 nm in absence of curcumin and 8 nm in presence of curcumin. The interfibrillar stacking distance is determined as 24 nm and 20 nm for the Q32-HttEx1 without and with curcumin, with the difference largely explained by the change in the fibrillar thickness.

## Curcumin affects the cellular response to HttEx1 fibrils
As discussed in the introduction, the debate on the toxic potential of HttEx1 oligomers and fibrils is not settled, but contributions from both species to cell death are plausible. It has been shown that HttEx1 fibrils can elicit cellular stress response, and that different fibril polymorphs can have different levels of responses or toxicity[11,12,14]. This raises the possibility that the fibril polymorphs obtained upon curcumin inhibition might mitigate the cellular stress induced by the HttEx1 oligomers formed normally. To examine this question, we performed three independent assays: MTT assays, to assess the cell metabolic activity of the cells, which is directly linked to cellular fitness; CytoTox-Glo assays, to address loss of membrane integrity cause by cytotoxicity; and Calcein AM staining, to assess neuronal network integrity through the measurement of neurite length. We challenged mouse hippocampal HT22 and human dopaminergic LUHMES cells with increasing concentrations of pre-formed Q32-HttEx1 fibrils (5 to 25 µM). HT22 cells' response to Q32-HttEx1 fibrils was examined following 24, 48, and 72 h of treatment, performing MTT assays to evaluate mitochondrial dehydrogenase activity, which correlates with the metabolic capacity of the mitochondria (Fig. 3a). For these assays, we used the Q32-HttEx1 fibrils because of the higher expression yields of this variant, given that these assays required substantial amounts of aggregates. We note here that prior studies by ssNMR and other techniques have failed to detect a qualitative difference in molecular structure between Q32- and Q44-HttEx1 fibrils, a finding supported by the ssNMR and TEM data in this study[14,26,28,29]. The uninhibited HttEx1 fibrils displayed a dose-dependent decrease in cell metabolic or mitochondrial reductase activity in the analyzed periods of treatment (Fig. 3a). The highest concentration tested (25 µM) led to a significant reduction in cell metabolic activity after 24-h treatment ($0.75 \pm 0.07$; $p < 00.5$), while

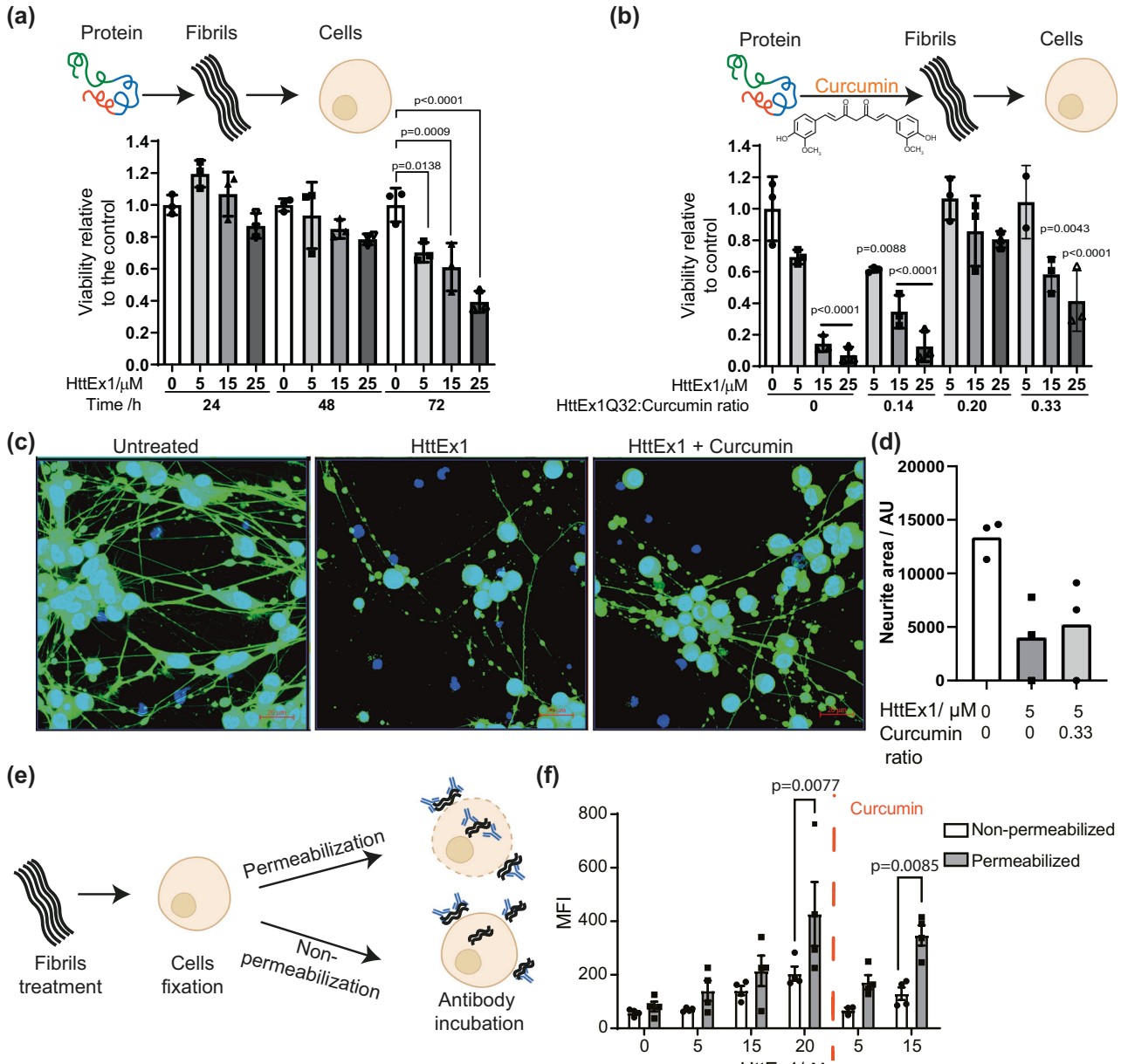

**Fig. 3 | Cellular response and cellular localization of Q32-HttEx1 fibrils prepared in the presence and absence of curcumin. a** MTT metabolic activity assay was performed following a 24, 48, and 72 h of exposure of HT22 cells with varying concentrations of HttEx1 fibrils. N = 3 independent experiments. One-way ANOVA followed by Bonferroni multiple comparisons test. Comparisons were performed against the untreated control for each incubation period. Bars represent the mean ± SD. Shading indicates protein concentration, as shown. **b** MTT metabolic activity assay was performed following a 72-h exposure of HT22 cells with varying concentrations of HttEx1 fibrils and HttEx1 fibrils prepared with different ratios of curcumin (as indicated. N = 3 independent experiments. One-way ANOVA followed by Dunnett's multiple comparisons. Indicated *p*-values corresponding to the comparison between the untreated control and the condition in which the value is

placed. Bars represent the mean ± SD. (**c**) Representative pictures of differentiated LUHMES cells exposed for 24 h to HttEx1 fibrcumin. Scale bars are 20 μm. **d** Total neurite length normalized by nuclei number. N = 1, with three experimental replicates. Shading indicates curcumin ratio, as shown. **e** Schematic representation of the experimental workflow for the determination of HttEx1 fibrils' cellular localization. **f** Detection of HttEx1 C-terminal His-tag after post-fixing permeabilization of cells treated with HttEx1 prepared in the presence or absence of curcumin. N = 4 independent experiments. Two-way ANOVA followed by Bonferroni's multiple comparisons test. Comparisons were performed between the non-permeabilized and permeabilized conditions for each treatment. Bars represent the mean ± SD. Schematic representations of the experiments were created using images from Servier Medical Art (Servier, http://smart.servier.com).

all concentrations tested displayed a significant decrease after 72 h of treatment. Mitochondrial reductase activity is directly associated with mitochondrial metabolic capacity, reflecting cell fitness and influencing cell response to stress. To evaluate if the MTT results reflected a direct increase in cytotoxicity, we measured the activity of cytosolic proteases released due to compromised membrane integrity, an indication of dying cells after 72 h treatment using the CytoTox-Glo assay (Supplementary Fig. 3). Our results show a

decrease in live cell activity with increased Q32-HttEx1 fibrils concentration (Supplementary Fig. 3a), indicating that the HttEx1 fibrils are cytotoxic.

Next, we evaluated the effect of the presence of curcumin during fibril formation (at sub stoichiometric molar ratios 0.14 to 0.33 of HttEx1: Curcumin) on the cellular response to these fibrils. Cells treated with fibrils formed in the presence of curcumin were able to preserve their cell metabolic activity better, displaying a mitigated

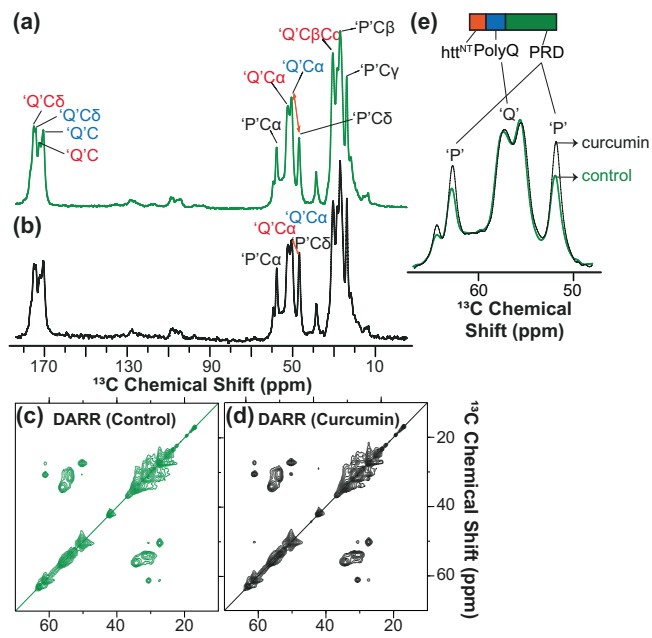

**Fig. 4 | MAS ssNMR spectra of U–$^{13}$C, $^{15}$N-labeled Q32-HttEx1 fibrils. a** 1D $^{13}$C CP spectra of fibrils formed at room temperature without curcumin (green), and **b** with curcumin (at the protein: curcumin ratio of 1:0.33) (black). The spinning frequency was 10 kHz. **c** 2D $^{13}$C –$^{13}$C DARR spectrum for U–$^{13}$C, $^{15}$N Q32-HttEx1 fibrils obtained at 13 kHz MAS and 25 ms of DARR mixing. **d** 2D $^{13}$C –$^{13}$C DARR spectrum for U–$^{13}$C, $^{15}$N Q32-HttEx1fibrils prepared in presence of curcumin (1:0.33) obtained at 13 kHz MAS and 25 ms of DARR mixing. **e** Zoomed-in overlay of 1D spectral regions showing Gln Cα peaks (middle) and Pro peaks, from spectra **a**, **b**. The NMR measurements were performed at 275 K on a 600 MHz spectrometer.

reduction in mitochondrial reductase activity after 72 h treatment in a manner that is dependent on the curcumin concentration used during the aggregation reaction (Fig. 3b). The CytoTox-Glo assay revealed a subtle reduction in the fibrils cytotoxic effect in HT22 cells, shown in Supplementary Fig. 3b and c. Noteworthy, treatment with curcumin alone or pre-treatment with curcumin prior to HttEx1 fibril treatment did not significantly alter the observed viability (Supplementary Fig. 4a).

As another measure of cytotoxic effects, we also compared the effects on neuronal network integrity of differentiated human dopaminergic cells, upon treatment with 5 μM Q32-HttEx1 fibrils formed in the presence and absence of curcumin during fibril formation (at a 0.33 ratio) (Fig. 3c, d). The deterioration in network activity is evident by the reduction of the neurite area observed in treated human differentiated dopaminergic LUHMES neurons compared to non-treated controls. The group exposed to fibrils formed upon curcumin inhibition did not show a significant improvement in the neurite area. Nevertheless, there was a higher number of cell bodies positively stained with Calcein AM (Supplementary Fig. 4b), which could reflect an improved viability, when compared to the group treated with fibrils formed in the absence of curcumin. Taken together, we observe that the mature HttEx1 fibrils exhibit toxic effects on both neuronal cell types, and that the cellular stress induced by the fibrils is mitigated by the effect of curcumin on fibril formation.

It is worth noting that no potentially disruptive techniques or compounds were applied to pro-actively facilitate the fibril uptake into cells, unlike certain prior studies that used e.g. lipofectamine[11]. This raises the question of whether the fibrils exerted their toxic effect due to uptake, and whether observed reductions (upon curcumin inhibition) in cellular stress may stem from reduced fibril uptake. To address these questions, we probed the subcellular localization by flow cytometry, based on detection of the His-tag attached to the C-terminal tail of the protein (Fig. 3e, f; Supplementary Fig. 5). MAS ssNMR shows this tail to be flexible and exposed on the surface of these HttEx1 fibrils (see below)[14,28–30]. Cells were treated with 5, 15 or 20 μM Q32-HttEx1 fibrils formed in the absence or presence of curcumin (at a 0.33 molar ratio) for 24 h. After treatment, cells were collected and fixed and each condition was divided in two samples, A and B. To distinguish between intracellular and extracellular fibrils bound to the cell membrane, we directly incubated sample A with the anti-His antibody-restricting the presence of the antibody to the extracellular environment, while cells in sample B were permeabilized to allow the anti-His antibody to detect fibrils that have entered the cells (Fig. 3e, f). Our findings indicate that fibrils formed in both the presence and absence of curcumin were detected on the outer surface of cells under non-permeabilized conditions, consistent with previous reports on the presence of Htt fibrils on the cellular surface[55]. When the cells were permeabilized we observed an increase in the His-tag signal, which implies that a portion of the fibrils was also found *inside* the neuronal cells. Importantly, this is true for both types of fibrils (formed with and without curcumin inhibition), with no sign of a decreased uptake of the fibrils formed under influence of curcumin.

## Curcumin changes the structure and dynamics of the fibrils

Thus, we have observed that curcumin inhibition results in fibrils that change in EM-observed morphology and cellular effects. In TEM and SAXS we observed not only changes in clustering behavior of the fibrils, but also a reduction in the width of individual filaments. We interpreted this as a change in the width of the fibril core, which is a characteristic feature of HttEx1 polymorphs[19,27]. To probe for molecular changes in the fibrils, we prepared uniformly $^{13}$C,$^{15}$N-labeled protein fibrils and examined them by ssNMR, using experiments previously used to distinguish fibril polymorphs by differences in structure and dynamics[12,14,30]. We measured ssNMR spectra of fibrils formed by U-$^{13}$C,$^{15}$N Q32-HttEx1 in absence and presence of curcumin (protein to curcumin ratio, 1:0.33), see Fig. 4 and Supplementary Figs. 6–7. 1D $^{13}$C cross-polarization (CP) ssNMR experiments show signals from rigid parts of the fibrils[56]. Figure 4a, b shows the $^{13}$C CP spectrum for fibrils grown in absence (green) and presence of curcumin (black). The obtained spectra closely match those seen previously for HttEx1 fibrils with a variety of polyQ lengths[14,19,26,28–30]. Comparing the two samples reveals a very high degree of similarity, both in terms of peak positions (chemical shifts) and peak intensities. This conclusion was reinforced by CP-based 2D ssNMR spectra ($^{13}$C-$^{13}$C DARR; Fig. 4c, d) showing the same polyQ fibril core signals. These observations immediately reveal that the atomic-level architecture within the buried core of the fibrils must be very similar, despite the fibril width differences seen by TEM and SAXS. Thus, while the width of the core changes, the ssNMR peaks match those of untreated fibrils and prior ssNMR studies of other polyQ aggregates[27,28,31,57].

The 1D and 2D spectra also contain other peaks not from the polyQ, in line with prior studies of HttEx1 fibrils[14,19,28–30]. Most notable are peaks from the proline residues that make up a large part of HttEx1, forming most of the C-terminal PRD. This HttEx1 segment is outside the rigid fibril core, displaying an increased degree of motion that is dependent on the fibril polymorph[12,14]. Interestingly, the proline peaks for fibrils made in presence of curcumin have a higher CP ssNMR signal intensity, compared to uninhibited fibrils (Fig. 4e; Supplementary Fig. 8). This shows that the PRD for the curcumin-modulated fibrils is more rigid than usual. We can observe this as a change in the relative heights of the polyQ core and PRD signals (Q/P signal ratio; orange arrows in Fig. 4a, b). In a separate batch of fibrils, we see a similar change in the Q/P ratio, but it was less pronounced, suggesting a degree of variability in this ordering effect (Supplementary Figs. 6–8). Decreased PRD motion could indicate an increase in steric hindrance associated with interactions affecting the PRD segments (more below).

The above data suggest a high similarity in the fibril core structure (at least on the local atomic level; see below), with more notable, but moderate, effect on the fibril surface dynamics. To probe the latter in more detail we also measured 1D and 2D ssNMR experiments based on the INEPT experiment, which selects for highly flexible residues (Supplementary Figs. 6, 7, 9). Also here fibrils prepared in absence and presence of curcumin were highly similar to each other, and closely resembled previously studied HttEx1 fibrils[14,19,27]. Peak positions (chemical shifts) appear essentially identical, but variations in peak intensity are observed. These findings also point to a difference in dynamics of the (C-terminal) flanking domains, dependent on curcumin inhibition. This observation is reminiscent of prior HttEx1 polymorphism studies by ssNMR, which have revealed that the HttEx1 polymorphs are mostly reflected in changes in the dynamics and interactions among the surface-facing flanking domains of the fibrils' fuzzy coat[12,14].

To test whether some of these changes may be a consequence of curcumin restructuring already formed fibrils, we also examined the effect of curcumin on pre-formed U-$^{13}$C,$^{15}$N Q32-HttEx1 fibrils (protein to curcumin ratio, 1:0.33). We performed TEM on these samples at different time intervals, which did not show a large change in morphology (Supplementary Fig. 10). NMR analysis of the post-treated fibrils as a function of time is shown in Supplementary Fig. 9a–j. 1D $^{13}$C CP experiments showed no significant changes affecting the polyQ core, further arguing against a disaggregation or restructuring effect. No significant changes in the peak intensity and peak position of the residues from PRD and htt$^{NT}$ were observed (Supplementary Fig. 10f–j). Only a small effect on the flexible C-terminal end of the protein was detected in INEPT spectra (Supplementary Fig. 10a–e). The limited impact of curcumin in these measurements is consistent with the compound primarily binding to the polyQ surface of the mature HttEx1 fibrils. These data also show that the fibrillar changes seen upon pre-treatment during aggregation are distinct from any effect of curcumin binding on fully formed fibrils.

## Discussion

We used curcumin as a small molecule inhibitor and studied its effect on the aggregation kinetics of HttEx1 protein. A clear inhibitory effect was observed that was present even for sub-stoichiometric amounts of curcumin. Mechanistically, the dominant effect was on the lag phase, which points to curcumin targeting the nucleation phase of the aggregation process[52,58]. This is noteworthy as nucleation processes are relevant as possible drivers of cellular protein aggregation and disease onset in HD[59]. Similar effects of curcumin have been seen with other amyloidogenic polypeptides, with it inhibiting Aβ fibril formation in vitro[45], and Aβ oligomerization and aggregation in vivo[46]. Under the sub-stoichiometric conditions studied here, we see that aggregation was significantly delayed, but that break-through aggregation into fibrils was nonetheless observed. Although not immediately apparent from fluorescence or kinetics analysis, an important finding related to the molecular structure of these break-through fibrils. The TEM, SAXS, and ssNMR analyses pointed to changes in the molecular conformation of HttEx1 fibrils, which depended on the curcumin concentration (Fig. 2). We will discuss the nature of the morphological change below, but first address the remarkable impact on the observed cellular response to the fibrils. We treated HT22 neuronal cells with HttEx1 fibrils formed in presence and absence of curcumin. We observed that both types of fibrils induced progressive reductions in mitochondrial reductase activity, overall cellular activity and neuronal network integrity (Fig. 3 and Supplementary Fig. 3b), similar to prior studies showing polyQ and HttEx1 aggregates to be cytotoxic[11,12,14]. However, the fibrils formed in presence of curcumin had less of an effect compared to the fibrils formed in absence of curcumin when analyzing the mitochondrial reductase activity, an indicative of cellular fitness[60]. So, the perturbing effect of curcumin on the HttEx1 aggregation process

not only delayed aggregation, but also mitigated the mitochondrial stress response triggered by the fibrils. However, only subtle changes were observed when evaluating cell death and network integrity. Taken together, our data suggest that cellular fitness in response to fibrils-induced stress is improved upon curcumin presence during fibril formation. It is important to note that fibril polymorphism is well-known to be context and condition dependent, such that the extent of this effect may be different under other experimental conditions.

It is worth exploring possible alternative contributions to the observed changes in cellular responses. We focused here on the impact of curcumin during the aggregation process in vitro, and specifically on the resulting fibril structure. This contrasts with studies of the effect of curcumin directly being administered to cells. The data in Fig. 3b were obtained by recovering fibrils formed in presence of curcumin, washing these fibrils, and adding them to cultured cells without the further addition of (or treatment with) curcumin. As such we attribute the observed rescue effect primarily to the observed differences in fibril structure. Yet, it is worth noting that our fluorescence assays showed that curcumin not only modulated aggregation but also showed a strong affinity for the formed fibrils (Supplementary Fig. 1). As such, the recovered fibrils had curcumin bound to them, raising the possibility of a beneficial effect of the bound curcumin. To understand whether the release of bound curcumin may underpin the observed rescue effect, we also treated cells with different concentration of curcumin in solution. Supplementary Fig. 4 shows that curcumin alone cause no change the cell viability percentage of the cells, thus confirming that the rescue effect is most likely explained by the changed structure of the fibrils.

### Structural changes in the HttEx1 fibrils

Our structural studies yielded data on the morphology and structure of the fibrils, pointing to changes as a consequence of the misfolding modulation by curcumin. The above-mentioned results may seem to point in different directions, but the employed techniques give insights on different microscopic levels: TEM having a resolution at the nm-level or above, whilst ssNMR gives insights on the sub-nm (Å) level. The TEM showed clear changes in the fibril diameter, which we attributed to a substantial structural rearrangement of the polyQ segment that forms the fibril core. A fibril width change was also detected and measured in the SAXS measurements, which also indicated a reduced radius for the fibrils made in presence of curcumin. On the other hand, the ssNMR showed that the polyQ core signals were unchanged, with no evidence of changes on the atomic level. How can these observations be reconciled?

We will now analyze and discuss the differences in morphology of these HttEx1 fibrils. The ssNMR data showed that the atomic-level structure within the polyQ core was analogous to that in our prior studies. There, we observed that the polyQ core of Q44-HttEx1 fibrils contained long extended β-strands, featuring up to 20 Gln residues per strand[15,19,23,57]. This corresponds to a β-strand length of 6-7 nm (20 Gln), reflecting ~3.5 Å for each β-strand residue. These extended β-strands stand apart from other amyloid fibrils, which often have short β-strand segments, as discussed in more detail in our prior work[19]. Figure 5a shows our published model for two polymorphs of Q44-HttEx1 fibrils[19], having average widths of approximately 6 nm (narrow) or 15 nm (wide). These width values represent an analogous TEM analysis as was performed here. As discussed previously[27], the polyQ fibril core is very rigid and dense, even compared to other amyloids, and as such impenetrable for the TEM stain molecules. Yet, these small molecules can penetrate between the flexible and disordered (seen by ssNMR) flanking segments on the fibril surface, which is not always the case[61]. As a consequence, we interpreted the fibril width seen by TEM (Fig. 2) to reflect the width of the fibril (polyQ) core, without contributions from surface-facing flanking domains. That said, a very precise interpretation of these dimensions may be limited by the expected[26] and

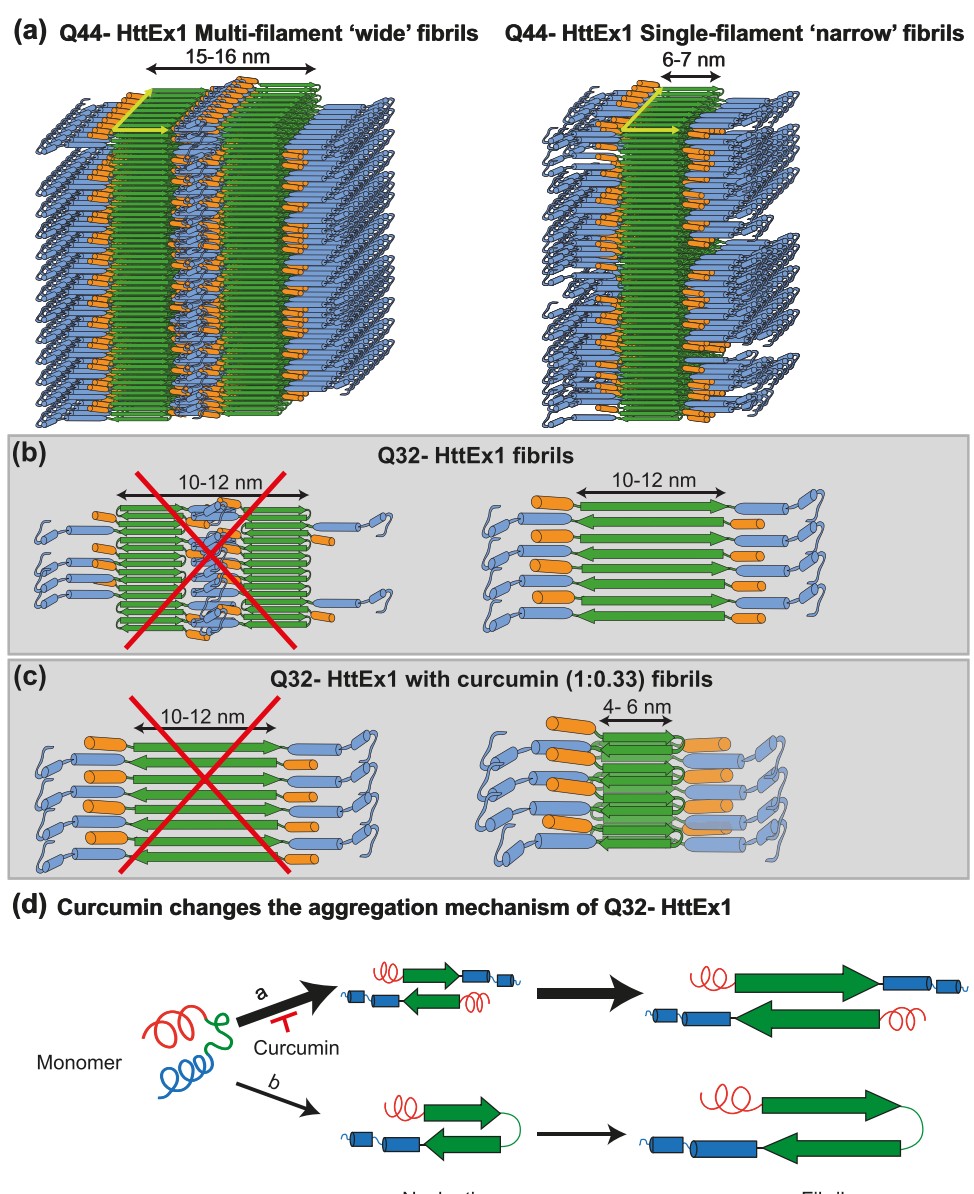

**Fig. 5 | Models for HttEx1 fibril structure and assembly. a** Existing models of Q44-HttEx1 multi-filament 'wide' fibrils (left) and a single-filament 'narrow' fibrils (right)[19]. Both models show a β-sheet core formed by polyQ segments in a β-hairpin fold. **b** Hypothetical models of Q32-HttEx1 fibrils formed in absence of curcumin, which match the observed TEM core width: double-filament fibrils with β-hairpins (left) or single-filament fibrils without (right). The ssNMR data show flexible PRD which is not consistent with the former model. **c** Hypothetical models of Q32-HttEx1 fibrils formed in presence of curcumin. Left model assumes an extended β-strand (no β-hairpin), but the predicted core diameter does not match TEM width. The model on the right shows the polyQ segment as a single filament featuring β-hairpins. **d** Schematic of the complex HttEx1 aggregation pathway, with competing mechanisms without (top) and with β-hairpin formation (bottom). The former pathway is dominant in absence of curcumin, but is inhibited by curcumin. Observed 'break-through' fibril formation (upon curcumin inhibition) results in fibril cores with a narrower width, consistent with the presence of β-hairpins. Panel a, b was adapted from ref. 19 under its CC open-access license.

observed[15] variations in the polyQ core architecture even within individual fibrils.

Building on our prior models[19], we might propose for the Q32-HttEx1 fibrils formed without curcumin, two models (Fig. 5b) that reflect the expected core width based on a 32-residues-long polyQ segment. To account for the TEM-measured 10-nm width, a model with β-hairpins (as seen for Q44-HttEx1) would require a double-filament model, given that the hairpin would yield single-filament width of ~5 nm. The TEM data could be consistent with this model (i.e., observed width 10-12 nm), however the ssNMR data do not show the reduced proline dynamics (due to inter-filament PRD interactions) expected from prior studies[14]. Those studies showed a large restriction of PRD motion in presence of

multi-filament interactions[12,14]. This should be apparent in the dynamics-sensitive CP- and INEPT-based spectra[56], but was not observed here. Thus, this suggests an alternative tertiary/quaternary structure, not based on a predominance of β-hairpins as seen in the Q44-HttEx1 fibrils.

Prior studies have supported a model in which β-hairpin-mediated polyQ aggregation is only dominant for long polyQ segments[21,27,62]. Shorter polyQ segments can and do aggregate, but were reported to depend on a multimeric nucleation process that is expected to result in fibrils in which protein monomers form extended β-strands without turn structures. This is illustrated in Fig. 5b (right); an implication of such an arrangement is that the N- and C-terminal flanking domains appear on either side of the core. In the absence of further filament-

filament interactions, these flanking domains are solvent exposed and expected to be flexible, as observed by ssNMR (INEPT spectra). This model also explains the observed TEM fibril core widths, and thus best fits the data observed for Q32-HttEx1 aggregated in absence of curcumin.

In presence of curcumin, we saw a clear reduction in the fibril width, reaching widths as low as 4-6 nm. This dimension is not compatible with the whole polyQ domain forming a single extended β-strand, as that would imply a core width of 10-12 nm. Thus, we conclude that there must now be a turn in the polyQ domain, such that it fits in a 4-6 nm core size. At this time, the abovementioned turn might be considered a β-turn (yielding a β-hairpin structure) or a β-arc type turn (which give a β-arch instead of a true β-hairpin)[63–65]. Based on prior support for β-hairpin formation in polyQ aggregation[21,27,62,66], we favor the former (a β-hairpin stabilizing β-turn), but we cannot exclude the possibility of other turn structures[63]. This leads us to the β-hairpin-based model in Fig. 5c (right), which would explain our EM data, with β-strands with approximately 14 glutamines for a length of 4.9 nm. As noted in prior work[15,19,23], the disposition of the flanking segments in such fibril architectures can vary and be harder to predict. We observed a decrease in PRD flanking segment dynamics by ssNMR. This could be explained by increased steric hinderance on the surface of the thinner fibrils, suggesting a dense local clustering of the PRDs. Another contributing factor could come from the increased long-range fibril-fibril clustering seen by SAXS and TEM, as this may be mediated by the C-terminal flanking segments.

## Mechanistic underpinnings of curcumin activity

We have shown that curcumin perturbs the nucleation processes in the aggregation mechanism, and that this sufficiently changes the HttEx1 aggregation pathway to yield different fibril polymorphs with different effects on cells. We also found that treatment of already formed fibrils was unable to achieve the same phenomenon. In Fig. 5d, we propose a two-pathway model to illustrate the role of curcumin in altering the aggregation pathway of Q32-HttEx1. The complexity of the HttEx1 aggregation pathway, with its parallel and competing misfolding processes, has been emphasized in prior work[21,31,62,67]. Our simplified schematic focuses on the role and fate of the polyQ segment itself, but does do so in context of HttEx1, where the flanking segments change the behavior of the polyQ segment (compared to polyQ model peptides). In our schematic the Q32 polyQ segment in HttEx1 favors an aggregation process that does not depend on intramolecular β-hairpin formation, yielding ultimately fibrils with an anti-parallel β-sheet core without β-hairpins. Prior experimental and computational studies suggest that β-hairpin formation as a nucleation event becomes the dominant pathway during aggregation of long polyQ, with aggregation by short polyQ being slower and occurring without β-turn formation[62,64,65]. Our data suggest that (in absence of curcumin) the dominant (i.e., fastest) aggregation process for Q32-HttEx1 may involve non-β-hairpin aggregation, explaining the wide-core fibrils. In contrast, when curcumin is present, this pathway may be disfavored leading to a slowing down of aggregation, while also leading to formation of thinner fibrils. The latter's width implies that they must feature short polyQ strands interspersed with turns (which we proposed to be β-turns but could also involve or include other turn motifs).

At first glance the favoring of turns (and/or β-hairpins) upon inhibition may seem to contradict the argument that β-hairpin-based aggregation should be the faster process. However, even if β-hairpin-formation is fast(er) than multimeric nucleation *for long polyQ*, the reverse may be true in short polyQ[62,68]. Here, our results suggest that curcumin impacts the misfolding process in such a way that a normally-faster aggregation process is disfavored, resulting in the normally-slower β-hairpin-based aggregation to become dominant. This could reflect a disfavoring of the former process (multimeric

nucleation), or a favoring of the latter process (polyQ turn formation), or a combination of both effects. Here, the findings for the Q44-HttEx1 protein may offer additional insight: also there, we saw a shift to narrower fibrils, akin to the effect on Q32-HttEx1. This could reflect a switch from a fibril architecture with a single turn in the polyQ segment (observed previously) to one where multiple turns are present (conceivably a combination of β-turns and β-arcs). This would be consistent with the curcumin acting by favoring the formation of turn motifs. That said, the change in Q44-HttEx1 fibril width (seen by EM) could also be explained by e.g. a switch from a double-filament architecture (Fig. 5a) to a single-filament one (Fig. 5b).

Thus, from our current data the precise molecular mechanism behind *how* curcumin modulates the misfolding process remains uncertain. A big challenge is the innate complexity of the natural HttEx1 misfolding process, which is not yet fully understood. On the one hand, it is conceivable that curcumin acts via direct interactions with the polyQ domain, (de)stabilizing certain conformations in the soluble monomeric protein. Alternatively, it may primarily interact with oligomers, modulating their structure or formation. On the other hand, also the HttEx1 flanking domains play key roles in the aggregation process. The N-terminal Htt[NT] segment is famous for driving oligomer formation and boosting aggregation kinetics, in particular at lower concentrations[31,67,69]. Thus, it also seems plausible that the effect of (hydrophobic) curcumin could be (in part) explained by interactions with flanking segments, e.g. amphipathic Htt[NT] α-helices. These mechanistic details warrant and require further investigation.

## Modulation of the fibrils' impacts on cells

The cellular stress response precedes cytotoxicity and can comprise a variety of pathways, including ER-stress response and autophagy, and directly affects the outcome – survival or cell death. This response depends on the stimuli and also on the cellular type and fitness status[70]. Our study further reinforces the findings from prior work that HttEx1 fibrils (made in vitro) have pathogenic effects on various types of cultured neuronal cells in multiple different assays. At the same time, our results add to growing literature that emphasizes that different types of HttEx1 fibrils display different degrees of cellular toxicity or metabolic disruption[11,12,14]. In HT22 mouse hippocampal cells, an immortalized and robust cell line, we observe a reduction in the mitochondrial metabolic activity upon the treatment with HttEx1 fibrils which is partially mitigated by the presence of curcumin during fibril formation. A similar, although more subtle, pattern is observed in the assays of cytotoxicity-associated membrane leakage. LUHMES dopaminergic human neurons, on the other hand, are more sensitive to the treatment. At 24 h, although there is no difference in neurite length upon treatment with fibrils formed in the absence or presence of curcumin, we observe that cell bodies are still positive for Calcein AM staining. This suggests that both cell types are more resistant to HttEx1 fibrils formed in the presence of curcumin, an effect that is more evident in long term (>24 h) response to exposure. The thinner fibrils made in the presence of curcumin triggered a mitigated cellular response to stress compared to the wider fibrils formed in absence of the inhibitor. As discussed, we attribute these differences primarily to a difference in fibril structure rather than the curcumin itself (Supplementary Fig. 3, Supplementary Fig. 4). It is worth noting that a priori we may have expected thinner fibrils to be more fragile and therefore more prone to cellular uptake and stress induction. Yet, our data may support the former (see cytometry data; Fig. 3e), but not the latter. In HD, proteinaceous inclusions form intracellularly, whilst our assays were based on extracellular administration of preformed fibrils. This experimental design was informed by our desire to control and analyze the structure of the fibrils in detail, and it also is a common strategy[10–12,14]. Moreover, extracellular aggregates may be relevant to HD in context of cell-to-cell prion-like propagation processes[4,71], while

mutant protein fragments (including aggregates) have also been detected in CSF[4,8,72].

An alternative explanation may be that the pathogenic properties of fibrils would be dictated (in part) by their surface properties[73,74], which define their 'interactome' in the cell. It is important to note that the fibrillar interactome does not only include (essential) proteins, but also other cellular components including membranes and lipids. Here, the ssNMR data are of particular value, as they provide a direct perspective on the fibril surface[14,25,75]. Dynamic flanking segments decorate the fibril surface, forming a 'fuzzy coat' implicated in protein-protein and protein-lipid interactions[76–79]. Therefore, a restructuring of the fibrils, which is seen to change the PRD dynamics, can change these cellular interactions. We propose that this may play a role in the modulation of cellular impacts of the curcumin-inhibited HttEx1 fibrils.

The targeting of the protein aggregation process itself is a potential strategy for treating HD[80], especially also with small molecules that delay or inhibit protein aggregation[17,24,38]. That said, the focus of the current work is on the use of curcumin as a model compound to evaluate possible (unexpected) side-effects of inhibition on the misfolding process, rather than advocating the actual use of curcumin as a direct treatment of HD[43]. Our data suggest that certain compounds might have a multifaceted beneficial effect, by not only delaying aggregation but also reducing the disruptive effects of any break-through aggregates on cellular metabolism and even viability. An important caveat of this finding is that one may also observe the opposite phenomenon, depending on the exact mechanism of inhibitory action and (cellular) context. It seems plausible that under certain conditions new polymorphs could be formed that display increased rather than decreased pathogenic or toxic activity. This is similarly possible when translating in vitro studies (as done here) to the aggregation of HttEx1 in cells or in vivo. These considerations should be kept in mind in such studies and warrant further investigation.

## Methods

### Protein expression and purification

The maltose-binding domain (MBP)-fusion proteins MBP-Q32-HttEx1 and MBP-Q44-HttEx1, featuring HttEx1 with 32 and 44 consecutive glutamine residues within the polyQ domain, was sub-cloned into pMALc5x and pMALc2x plasmids, respectively, by Genscript (Piscataway, NJ)[19]. The fusion protein was overexpressed in E. coli BL21(DE3) pLysS cells (Invitrogen, Grand Island, NY). The cells were grown in 2 L LB medium with ampicillin and chloramphenicol at 37 °C and 200 r.p.m. until an optical density (OD600) of 0.80. A temperature ramp from 37 °C to 18 °C over 30 mins at 200 r.p.m. was applied to prepare for induction. Proteins for study by ssNMR were uniformly $^{13}$C, $^{15}$N labeled by growing the bacteria on M9 minimal media supplemented with $^{13}$C-D-glucose and $^{15}$N ammonium chloride (Sigma-Aldrich) during overexpression[14]. Protein expression was induced by adding 0.6 mM IPTG (Sigma-Aldrich). The protein was overexpressed at 18 °C for 16 h, after which the cells were pelleted at 5250xg for 20 mins. The cells were resuspended in Phosphate Buffered Saline (PBS) buffer (11.9 mM phosphate, 137 mM sodium chloride and 2.7 mM potassium chloride), pH 7.4. The resuspended cells were kept on ice, followed by the addition of 1 mM phenylmethanesulfonyl fluoride (Sigma-Aldrich), and 0.5 mg/ml of lysozyme (Sigma-Aldrich). The cells were burst using sonication (VCX130 Vibra-cell sonicator, Sonics & Materials, Inc), applying 70% amplitude for 20 mins with alternation of 10 s pulses and breaks of 10 s. Cell debris was removed by centrifuging at 125,000xg for 45 mins. The soluble protein was filtered using 0.45 μm syringe filters and purified using a HisTrap HP nickel column (GE Healthcare) with an imidazole gradient. The purified protein was exchanged into an imidazole-free PBS buffer (pH 7.4) using dialysis membrane of 12400 MWCO (Sigma-Aldrich). The concentration of fusion protein was determined

by its average absorbance ($n = 3$) at 280 nm, as measured in a 50 μl quartz cuvette with a 10 mm path length by a JASCO V-750 spectrophotometer. The extinction coefficient of the fusion protein is estimated to be 66,350 M$^{-1}$cm$^{-1}$ (from the protein sequence by the ProtParam tool by ExPASy[81]). The protein purity, labeling, and molecular weight were verified by ESI-TOF MS and SDS-PAGE (12%)[19].

### Fibril formation

The purified fused protein was cleaved in PBS buffer by treating it with Factor Xa (FXa) (Promega, Madison, WI) at room temperature to release the HttEx1 (HttEx1:FXa = 400:1 molar ratio). The progression of cleavage and aggregation were monitored by SDS-PAGE (Bio-Rad mini protein Precast TGX Gels 12%) and TEM. The fusion protein was cleaved and allowed to aggregate over 4 days, after which the HttEx1 fibrils were pelleted down at 3000xg for 20 min. The supernatant was discarded, and pelleted fibrils were washed at least three times with PBS buffer. For MAS NMR-analyzed samples, the fusion protein was used in a uniformly $^{13}$C,$^{15}$N-enriched form (samples 1-5 in Supplementary table 1).

For Sample 5, 71 μM of MBP-Q32-HttEx1 protein was cleaved by treating it with FXa at room temperature to release the HttEx1 (HttEx1:FXa = 400:1 molar ratio). The fusion protein was allowed to aggregate over 4 days, and then the fibrils were pelleted down at 3000xg for 20 mins. The fibrils were washed three times with PBS buffer and then re-suspended in PBS. Then, these pre-formed fibrils were treated with 23.6 μM of curcumin in PBS for a 1:0.33 protein to curcumin molar ratio. For the TEM studies, the sample was collected at specific time intervals (24, 48, 72 h). For ssNMR, these fibrils with curcumin were packed in the rotor, and the sample was measured at specific time intervals (24, 48, 72, 96 h).

### Solid-state NMR spectroscopy

All solid-state NMR experiments were acquired on a Bruker Avance NEO 600 MHz spectrometer, equipped with a 3.2 mm MAS probe with an HCN Efree coil (Bruker Biospin). Isotopically labeled Q32-HttEx1 fibrils in the absence and presence of curcumin were prepared and then packed by pelleting a hydrated suspension of purified protein fibrils into 3.2 mm zirconia thin wall MAS rotors (Bruker Biospin, Billerica, MA). This sedimentation process was done using a home-built ultracentrifugal packing device under centrifugation at ~130,000xg in a Beckman Coulter Optima LE-80K ultracentrifuge equipped with an SW-32 Ti rotor[82]. A summary of sample details is provided in Supplementary Table 1, referring to the individual preparations as Samples 1 through 5. Samples 1 and 2 were prepared together (referred to as Batch 1), being Q32-HttEx1 (control; Sample 1) and Q32-HttEx1 with curcumin (1:0.33). Samples 3 and 4 represent separate Batch 2, of otherwise identically treated Q32-HttEx1 (control) and Q32-HttEx1 with curcumin (1:0.33). In case of sample 5, the labeled Q32-HttEx1 fibrils were treated with curcumin (1:0.33) after preparation of the fibrils. Caps were sealed to the rotor with epoxy glue to ensure stable sample hydration. Samples were studied by MAS ssNMR in a hydrated and unfrozen state. All the 1D experiments for the Batch 1 and Batch 2 fibril samples were recorded with a spinning frequency of 10 kHz, 1024 scans, and at 275 K. The 2D $^{13}$C-$^{13}$C DARR[83] experiment was performed with a 25 ms mixing time with 40 scans at 13 kHz spinning rate. The 2D $^{13}$C-$^{13}$C INEPT- TOBSY (P9$^1_3$) experiment was performed at 8.33 kHz MAS with 32 scans[84]. More experimental details are in Supplementary Table 2. SSNMR peak integration analysis was done on the $^{13}$C cross-polarization spectra obtained for batch 1 and batch 2, Q32-HttEx1 (control) and Q32-HttEx1 with curcumin (1:0.33) fibrils. Spectra were acquired with Bruker Topspin v. 4.1.3, processed with NMRPipe v.11.5[85], analyzed in CCPNMR Analysis v.2.4. Peaks were integrated between the mentioned chemical shifts ranges with a python script (available from nmrglue website[86]). For batch 1 fibrils, the peak area for PCα was selected from 68.31 ppm to 60.68 ppm; for QCα - 60.68 ppm

to 53.40 ppm and for PCδ - 53.40 ppm to 47.78 ppm. For batch 2 fibrils, the peak area for PCα was selected from 66.70 ppm to 58.99 ppm; for QCα - 58.99 ppm to 51.79 ppm and for PCδ - 51.79 ppm to 46.01 ppm. The obtained values were normalized by dividing them by the maximum intensity corresponding to the QCα peak and visualized in Microsoft Excel. Error bars represent an estimate of the noise.

## ThT Assays

Unless noted otherwise, for aggregation kinetics analysis, 61 μM MBP-Q32-HttEx1 protein was used, with cleavage performed with FXa at a 400:1 molar ratio (HttEx1: FXa). ThT was included in the black polystyrene, clear bottom, 96-well plates (Corning, U.S.A) at a final concentration of 15 μM. A 1 mM Thioflavin T (ThT) stock was prepared in DMSO, which was subsequently diluted in PBS buffer. Curcumin stock was prepared (1 mM) in DMSO and diluted in PBS buffer for spectroscopy at a final concentration of 8.5 μM, 12 μM, and 20 μM. Samples were mixed gently and then measured on a TECAN Spark 10 M microplate reader. Excitation occurred at 442 nm and emission was recorded from 482-492 nm, with excitation and emission bandwidths of 10 nm. The plate was shaken for 5 s before recording the emission for each measurement. The emission was recorded every 15 minutes at 490 nm and was used for plotting the fluorescence kinetics curves (Fig. 2), based on the subtraction of the signal from the control (PBS buffer with ThT). Plotted data were normalized to the maximum observed signal. For studying the aggregation kinetics of Q44-HttEx1, the same protocol was followed with a HttEx1 protein concentration 67.5 μM and curcumin concentrations of 10 μM and 20 μM. The aggregation curves in Fig. 2a, b were plotted in Origin 8.1 software, and represent the average values of data measured in triplicate. The aggregation curves with error bars are shown in Supplementary Fig. 2a–c. Data measured in absence of curcumin (Supplementary Fig. 1a, b) were also analyzed with the AmyloFit program[87].

## Transmission Electron Microscopy

Transmission electron microscopy (TEM) was performed on mature Q32-HttEx1 and Q44-HttEx1 protein fibrils. 61 μM of the fused Q32-HttEx1 and 50 μM of Q44-HHttEx1 protein was cleaved by treating it with FXa (Promega, Madison, WI) at room temperature to release the HttEx1 (HttEx1:FXa = 400:1 molar ratio). The protein fibrils which were allowed to aggregate at room temperature for 96 h in the absence and presence of different concentrations of curcumin. For the studying the effect of curcumin on the 71 μM of the Q32-HttEx1 pre-formed fibrils, these fibrils were treated with 23.6 μM of curcumin (i.e. protein: curcumin; 1:0.033 molar ratio). For the imaging studies, the fibrils were suspended in MiliQ water. The fibril samples were deposited on plain carbon support film on 200 mesh copper grids (SKU FCF200-Cu-50, Electron Microscopy Sciences, Hatfield, PA) after glow-discharge for 0.5 -1 min. The excess MiliQ was removed by blotting. The negative staining agent used was 2% (w/v) uranyl acetate. The stain was applied immediately after blotting for 0.5 -1 min. The excess stain was removed by blotting and the grid air dried. The images were recorded on a Philips CM120 electron microscope operating at 120 kV. Images were recorded on a slow scan CCD camera (Gatan). The fibril widths were measured transverse to each fiber axis using the straight free-hand tool of Fiji[88]. Each measurement spanned the length of the negative stained area of the fibril with similar contrast The width measurements are assumed to reflect the polyQ core, as the stain accumulates on the flanking domains[19]. In images with low resolution, the fibril diameter was determined in regions with the clearest defined boundaries. Three measurements were obtained per fibril, except for fibrils where the width varied significantly. In selected cases, fibril widths were verified on isolated and vertically aligned fibrils using the Plot Profile tool of Fiji, which plots the average grayscale intensity values across the fibril axis.

## Cell Culture

Mouse HT-22 hippocampal cell lines and Lund human mesencephalic cells (LUHMES)- differentiated neuronal cells were kindly provided by Prof. Culmsee, University of Marburg, Germany. HT22 cells were cultured in Dulbecco's Modified Eagle Medium (Gibco, ThermoFisher Scientific, Landsmeer, The Netherlands) supplemented with 1% pyruvate (ThermoFisher Scientific), 10% fetal bovine serum (GE Healthcare Life Sciences, Eindhoven, the Netherlands), and 100 U/mL penicillin-streptomycin. LUHMES cells were cultured in Advanced DMEM/F12 (Gibco), supplemented with 200 mM L-Glutamine (Gibco), 1x N2 Supplement (Gibco), 100 μg/ml FGF (PeproTech, Cranbury, USA), and 1x penicillin/streptomycin solution (Gibco). LUHMES cells were differentiated to dopaminergic neurons before the treatments[89,90]. Cells were maintained at 37° and 5% $CO_2$. Cells were used for experiments for at most 10 passages after thawing and regularly checked for Mycoplasma infection.

## MTT assay

Cell viability was assessed using MTT (3-(4,5-dimethyl-2-thiazolyl)-2,5-diphenyl-2H-tetrazolium bromide; Sigma-Aldrich) reduction assay[91]. HT22 cells were seeded in p96 wells at a density of $9.10^3$ cells/well. After 24 h of seeding, the cells were treated with 0, 5, 15 or 25 μM of pre-formed Q32-HttEx1 fibrils diluted in culture medium and incubated for 24, 48 or 72 h. For endpoint analysis, cells were incubated with 0.5 mg/ mL MTT for 1 h at 37 °C and 5% $CO_2$. The supernatant was removed and the plate was stored at −20 °C for at least two h. Afterwards, 100 μL of dimethyl sulfoxide (DMSO; Sigma-Aldrich) was added to each well and the plate was incubated for 1 h under mild shaking at 37 °C. The absorbance of each well was determined at 570 nm with reference background absorbance at 630 nm using a Synergy H1 Multi-Mode reader (Biotek, Winooski, US). Absorption values were normalized to the average of the untreated control to determine cell viability in reference to the untreated control (reference value 1).

## Neuronal network integrity assay

LUHMES cells were seeded in pre-coated 48-well Nuclon TM plates at a density of $5.10^4$ cells/well. After 6 days of differentiation, the cells were treated with 5 μM of pre-formed HttEx1 fibrils, and incubated curcumin. The cells were treated for 24 h at 37 °C and 5% $CO_2$. Neuronal network integrity was determined after staining with Calcein-AM[92]. To do so, cells were stained with 1 μM Calcein-AM (Sigma-Aldrich) and 1 μg/mL Hoechst (Invitrogen, ThermoFisher) for live cell imaging. The neuronal network was imaged with the Zeiss Cell Discoverer 7 (Zeiss, Oberkochen, Germany) and analyzed with the Fiji Macro Plug-In NeurphologyJ Interactive, with neurite area as the dependent variable[93].

## Flow cytometry

To estimate the subcellular localization of the Q32-HttEx1 fibrils, the presence of the C-terminal His-tag of fibrillar Q32-HttEx1 was detected by flow cytometry. HT22 cells were treated with 5, 15 or 20 μM Q32-HttEx1 formed in the absence or presence of curcumin (at a 0.33 ratio) for 24 h. Cells were harvested with trypsin and permeabilized or not with 0.1% Triton-X (Sigma) in PBS for 15 min at RT. Next, cells were blocked with 1% BSA (Sigma) in PBS for 1 h, followed by incubation with 1:500 anti-THE™ His Tag (Cat# A00186S, GenScript, New Jersey, US) for 2 h at RT. Secondary antibody Alexa Fluor 488 nm Donkey anti-mouse, at a 1:2000 dilution, (Jackson, Cambridgeshire, UK; catalog number 715-545-020) was incubated for 2 h, RT. The cytometry gating strategy consisted of exclusion of debris (Supplementary Fig. 5). The cytometry data was collected with CytoExpert v2 and analyzed using FlowJo v9.0 and Kaluza analysis software v2.1. Fluorescence was measured at 690/50 nm in the CytoFLEX benchtop flow cytometer (Beckman Coulter Life Sciences, Indianapolis, US).

Analysis was performed using FlowJo v9.0 (Becton, Dickinson and Company, Franklin Lakes, US).

## Cell toxicity assay

Cell toxicity was assessed using the CytoTox-Glo™ Cytotoxicity Assay (Promega). HT22 cells were seeded in white-walled 96-well plates at a density of $8 \times 10^3$ cells per well in 100 μL of medium and incubated at 37 °C with 5% $CO_2$ for 24 h. The cells were then treated with 5, 15, or 25 μM of pre-formed Q32-HttEx1 fibrils in medium for 72 h, with untreated wells serving as negative controls. After treatment, 50 μL of CytoTox-Glo™ Reagent was added to each well. The plates were briefly mixed, incubated for 15 minutes at room temperature, and luminescence was measured to quantify cell death (read 1). Then, 50 μL of Lysis Reagent was added to lyse all cells, followed by another 15-minute incubation. Luminescence was measured again to determine total cell activity (read 2). Measurements were taken using a Synergy H1 Multi-Mode reader (Biotek, Winooski, USA). Live cell activity was determined according to the equation: Live Cell Activity = Total Luminescence (read2) − Experimental Dead Cell Luminescence (read1). The data was normalized by the average of the cell activity of the control group.

## Small angle X-ray measurements (SAXS)

63 μM of MBP-Q32-HttEx1 was cleaved using Factor Xa protease in molar ratio (FP:P; 400:1) and aggregated in presence and absence of curcumin (1:0.33) at room temperature. After 96 h, the fibrils were collected as described above. The fibrils were centrifuged at 2400xg, and the supernatant (PBS) was removed such that to make the final concentration of the fibrils to 2 mM for the SAXS experiment. The SAXS measurements were performed at the multipurpose X-ray instrument for nanostructured analysis (MINA) at the University of Groningen. The instrument is equipped with a high flux rotating anode source using radiation from a Cu anode and delivering X-ray photons with a wavelength λ = 0.15413 nm (E = 8 keV). The SAXS patterns have been acquired using a solid-state noiseless Pilatus 300k detector (Dectris) placed 3.1 m away from the sample. The coordinates of the transmitted X-ray beam and the angular range probed by the measurements were calibrated using a standard diver behgenate powder (NIST). 2D SAXS patterns were reduced to 1D SAXS profiles using Fit2D and Origin software, after proper correction for the difference in sample absorption and subtraction of the background (PBS buffer). Samples were measured in 1.5 mm glass capillaries at ambient temperature (23 °C). The SAXS curves are plotted as I(q) vs q in the log-log scale, where $q = (4\pi \sin\theta)/\lambda$ is the modulus of the scattering vector and 2θ is the scattering angle.

## SAXS data analysis

The SAXS curves have been analysed using the model for a concentrated ensemble of long rod-like objects characterised by length $L$ and radius $r$. The general equation for the fitted model in the framework of the monodisperse approximation is shown in Eq. 1:

$$I(q) = A\left[\int_0^\infty P(q,r,L)N(r)dr\right]S(q,d,\nu) + I_{bkg} \qquad (1)$$

Where $P(q,r,L)$ is the function called form factor, describing the scattering for the rod-like objects, and $S(q,d,\nu)$ is the function called the structure factor, describing the lateral packing of the rod-like objects, with average interfibrillar distance $d$ and interaction parameter $\nu$. The fibrillar cross-sectional dimension is subjected to a log-normal size distribution $N(r)$, characterised by an average dimension $r$ and width $\sigma$. The prefactor A is a constant which includes the number density of objects (i.e., concentration) and the squared of the average electron contrast between the Q32-HttEx1 fibrils and the surrounding media. The background intensity $I_{bkg} = \frac{B}{q^n} + C$ takes

into account all atoms and molecules which are not contained in the measured background and are not included in the assembled fibrillar domains.

As form factor we used the standard equation for long cylindrical objects[94]. Since the length $L$ of the rod-like objects is outside the probed range, its value was kept fix to 1000 nm during fitting. For the structure factor we used the expression derived for the Polymer Reference Interaction Site Model (PRISM)[95,96]. In this model, the interaction parameter ν is associated with the second virial coefficient describing the pair interaction between adjacent cylinders. All equations are implemented in the SASfit software[97] which was used to fit the SAXS 1D profiles. The fitted parameters are thus $A$, $r$, $d$, $\nu$, $B$, $C$ and $n$ (Supplementary Table 3).

In addition to the model above, the data have been fitted also using a model previously reported by Perevozchikova et al. for single huntingtin fibrils[54]. The average fibrillar dimensions obtained using this model and characterized by a radius of cross-sectional radius of $R_{g1}$ are in close agreement with the ones obtained by our model. The fitted curve using the model by Perevozchikova et al. for the Q32 sample prepared without curcumin is reported in Fig. 2o in the main manuscript as dashed line, while the values for the fitted parameters are reported in Supplementary Table 4 for fibrils prepared without and with curcumin. Supplementary Table 5 contains essential parameters related to the SAXS data acquisition and analysis.

## Reporting summary

Further information on research design is available in the Nature Portfolio Reporting Summary linked to this article.

## Data availability

Source data have been deposited in the Zenodo database and are accessible here: https://doi.org/10.5281/zenodo.14906545. Additional data are available from the corresponding authors upon request.

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

## Acknowledgements

We thank Dr. Marc Stuart for the Electron Microscopy facility. We also thank Dr. Alessia Lasorsa for her help with the solid-state NMR experiments, and Gea Schuurman-Wolters for her help with the wet-lab instruments. We acknowledge funding from The CampagneTeam Huntington and CHDI Foundation (contract A–17778).

## Author contributions

G.J., M.T.-L., I.M., G.P., H.T.R., T.C., and R.P. performed research; G.J., I.M. performed and analyzed ssNMR and participated in paper editing; G.J., M.T.-L., G.P., I.M. analyzed data; A.M.D., P.v.d.W., designed and supervised the research; G.J., M.T.-L., G.P., A.M.D, P.v.d.W wrote the paper.

## Competing interests

The authors declare no competing interests.
