## [Transparent Peer Review file · Nature Communications]

Inhibitor-based modulation of huntingtin aggregation mechanisms mitigates fibril-induced cellular stress

Corresponding Author: Professor Patrick van der Wel

Version 0:

Reviewer comments:

Reviewer #1

(Remarks to the Author)

In this contribution, van der Wel et al investigate the influence of Curcumin on the aggregation of the 32-residue long PolyQ containing construct Q32-Htt.Ex1 in terms of aggregation kinetics and structure, dynamics and cytotoxicity of the resulting fibrils.

The addition of curcumin inhibits the aggregation, with increasing amounts of curcumin result in an extended lag phase. Likewise, the average diameter is reduced for fibrils grown in the presence of curcumin. On the other hand, the conformation of the individual glutamine residues within the fibrils as probed by solid-state NMR-spectroscopy is not altered by the addition of curcumin. However, Proline signals from the neighboring, mostly flexible proline rich domain seem to be slightly more enhanced in dipolar-based spectra of curcumin-grown fibrils, pointing towards slightly reduced mobility of this domain. Finally, cytotoxicity towards a neuronal cell culture of Q32-Htt.Ex1 fibrils grown in presence of curcumin is reduced with respect to that on fibrils grown in the absence of curcumin, and most of the fibrils regardless whether grown in presence or absence of curcumin do not enter the cells.

These reduced dimensionality of curcumin-grown fibrils is explained by the hypothesis that addition of curcumin favors the formation of beta-hairpins, which slows down the aggregation of Q32-Htt.Ex1.

This article presents a sound study of the potential effects of a "fibrillation inhibitor" on the aggregation process of Q32-Htt.Ex1 *in vitro*. The authors do state very clearly that they have studied *in vitro* effects, that aggregation is only slowed down and not prevented, and that the observed of reduced toxicity of *in vitro* grown fibrils can not directly transferred to claims on *in vivo* effects.

Most results are well justified and hypotheses are clearly indicated as such. Only the conclusion that the Pro-rich domain is less flexible in fibrils grown in presence of curcumin should be handled with more care: at least in the second batch of fibrils this effect seems to be negligible. And in any case, a quantification (relative intensity of the integral of the Pro peak with respect to one or more significant Gln peak(s) would be helpful in this respect).

On a minor point: Figure legend 3a : to my understanding, Figure 3a shows only results on fibrils grown in absence of curcumin, so the second half of the sentence (and HttEx1 fibrils prepared in presence...) should be omitted.

Reviewer #2

(Remarks to the Author)

The basic flaw in this work is the outdated references to Huntington's disease, that are repeatedly either >25 years old or just wrong.

"Huntington's disease (HD) is a neurodegenerative disorder in which a mutated fragment (exon 1) of the huntingtin (HttEx1) protein undergoes misfolding and aggregation."

No, HD is not caused by protein misfolding and aggregation.

1. There is no correlation between aggregate presence and neuronal dysfunction in HD brains.
2. There is evidence from 2006 on that any aggregates may be protective in neurons.
3. In human iPSC derived neurons, and human derived model lines, there are clear HD phenotypes, and no protein inclusions.
4. In 2016 Human GWAS studies, now at over 16,000 patients, there is no evidence that any proteostasis pathways are modifiers of disease onset, nor severity.
5. There was one transgenic model of huntingtin Exon1 from 1996, for which no control strain exists. Unfortunately, this has become dogma because it follows an simplistic amyloid hypothesis for which there has been a plethora of contradictory

evidence in the last 15 years.

6. In symptomatic patient brains who died prematurely, there is little evidence of protein aggregate pathology.

The second problem is curcumin which has been funded and studied by the NIH for a decade or more and proven not to be bioavailable. I will politely refer to curcumin research in general as "controversial".

Huntingtin exon1 is not a model of Huntington disease. It's barely 3% of the protein that is now known to be a dimer with HAP40 in a complex of over 380KDa. Since 2018, groups have made huge quantities of Cryo-EM quality huntingtin and no one has ever reported the presence of aggregate. Even in Cryo-EM studies, the first 500 amino acids are seen as unstructured.

Biochemistry is chemistry in a living context. Chemistry is entirely reliant on proper stoichiometry, yet among amyloid researchers, the model context is typically, as in this work, several orders of magnitude above physiological stoichiometry, which implies that stoichiometry is irrelevant and why this type of research has gone nowhere in translation in 30 years.

Figure 2: there is no consideration of unbiased quantification, 2b is a single measure, somehow presented as a bar graph.

Figure 3: the use of Q32 exon1 model is strange, because this is not the pathogenic threshold of CAG allele expansion seen clinically at 37 or more, with a mean of 43. This just proves that massive overexpression of a synthetic fragment will cause an aggregation event that is irrelevant to human disease. No control of <Q21 wild type allele length.

3d: bar graphs with lower error bars removed for aesthetic reasons, not appropriate. Same figure T-test analysis on three conditions, which should be ANOVA. Then a complete different format appears for 3F. More care should be taken in figure and data presentation.

No mention of effects on exon1 PTMs, which are known to affect aggregation.

Overall, a very naïve study with little modern knowledge of the state of HD research using outdated model with no controls for biological relevance, landing on a compound that cannot be bioavailable.

Reviewer #3

(Remarks to the Author)

In this well-written article, Jain et al. provide compelling evidence that curcumin, a known inhibitor of amyloid formation, alters the mechanism of aggregation by huntingtin protein in vitro. In the presence of substoichiometric concentrations of curcumin, amyloids formed with delayed kinetics, an altered structure, and reduced toxicity. These findings will stimulate further work into the mechanism of potentially therapeutic small molecule-based modulation of amyloid formation.

Critiques:

1. In the abstract and introduction, it is stated that deposits/deposition/inclusions are toxic, while the authors elsewhere intend to be agnostic about the nature of toxic species. I would therefore replace these terms with something less specific, such as "aggregates" or "misfolded species".
2. Structural features of amyloids, such as the specific polymorph produced, tend to be extremely sensitive to concentration, temperature and other conditions of the amyloid assembly reaction. It is therefore desirable to provide some explanation of the conditions used for assembly, and rationale for choosing them, in the first results section.
3. The increased lag phase in the presence of curcumin is interpreted as evidence that curcumin inhibits primary nucleation. However, a qualitative analysis of ThT kinetics does not suffice to make this conclusion. As exhaustively explained by Linse, Knowles and colleagues (see e.g. <https://doi.org/10.1039/C4CP05563B>), the lag phase is a product of primary and secondary processes; it is not a simple indicator of primary nucleation rates. The curves shown here can, in principle, be fit to kinetic equations (e.g. AmyloFit) to distinguish the effects of curcumin on impacts primary or secondary processes.
4. I am dismayed that the authors focus on Q32 Htt throughout this work, while relegating data for Q44 Htt to the supplement. They do not provide a justification for this choice, even though the latter is uniquely relevant to disease. Likewise, in the discussion, they choose to interpret Q32 with respect to their prior findings with Q44, while oddly avoiding any mention of the present data on Q44.
5. Although the assay used to analyze toxicity is not atypical for the amyloid field, do we have any reason to believe that it assesses a disease-relevant mechanism? I.e. is there evidence that extracellular Htt aggregates contribute in any way in vivo? Is there known to be a correlation between extracellular and intracellular toxicity? If not, please clarify the relevance of results using this toxicity assay, even if just to further indicate that the fibrils are different.
6. Do the stated concentrations of fibrils used in the toxicity assay refer to the concentrations of resuspended fibrils as determined experimentally, or do they simply assume 100% conversion of starting protein? Please clarify if (and how) the concentration of fibrils was normalized to ensure that equal amounts of treated vs nontreated fibrils were compared. This would help in interpreting the reduced toxicity of curcumin-treated fibrils with respect to structure rather than a possible reduction in the extent of fibrillization. I do find that the subsequent quantitation of fibril binding/uptake by flow cytometry goes a long way to address this concern.

7. The fibril widths extracted from TEM data are assumed to reflect the widths of the amyloid cores. This assumption is inconsistent with published work showing that TEM over-estimates the widths of fibril cores (e.g. <https://doi.org/10.1038/s41593-023-01341-4>, <https://doi.org/10.1002/1873-3468.13675>). This is importance for the structural interpretations that rely on these measurements.

8. The interpretation of Q32 fibrils as lacking a hairpin is inconsistent with multiple prior studies showing that polyQ segments down to 26 residues in length aggregate via a β -hairpin nucleus. This was previously reviewed by Matlahov and Van der Wel 2019 (authors on the present paper). I do not consider fibril widths obtained from TEM sufficient to overturn these prior studies. If one accounts for the likelihood that the stain does not fully penetrate to the core, then the core will be smaller than the observed width, and is not inconsistent with a turn-containing model.

9. Likewise, the explanation for why Q32 may disfavor a hairpin seems inconsistent with prior in vitro and in silico observations that polyQ in this length regime is a highly collapsed globule (e.g. <https://doi.org/10.1073/pnas.0608175103>, <https://doi.org/10.1073/pnas.0900678106>, <https://doi.org/10.1016/j.jmb.2009.08.034>) driven by intramolecular H-bonding. It is therefore not necessarily true that a hairpin would be entropically disfavored relative to an extended strand (I would argue the opposite). In any case, it will depend on whether the extended strand can be stabilized intermolecularly and therefore depend on concentration.

10. Though not discussed, the median fibril width for Q44 treated with curcumin is between 3 and 4 nm as shown in Supp Fig 2. This width suggests more than one turn and/or arc per monomer, consistent with the nucleus model of Kandola et al. 2023 (<https://doi.org/10.7554/eLife.86939.1>) for polyQ >36. The effect of curcumin could therefore be interpreted as inhibiting the transition from this short-stranded core to the longer single hairpin core of mature polyQ amyloid. Stated differently, the data appear to be consistent with curcumin stabilizing an arch-containing polymorph that is otherwise only transient. Assuming this species grows slower than the mature amyloid polymorph, this explanation is also consistent with the slower kinetics in the presence of curcumin.

Version 1:

Reviewer comments:

Reviewer #1

(Remarks to the Author)

The authors have addressed all my concerns and those of the other two reviewers. Thus, the manuscript is in my point of view ready for acceptance.

Reviewer #3

(Remarks to the Author)

This revised version of Jain et al. adequately addresses my critiques of the original submission, and is appropriate for publication. I commend the authors on their thoughtful, professional, and very thorough handling of the reviewers' comments.

Reviewer #4

(Remarks to the Author)

Jain et al.

Inhibitor-based modulation of huntingtin aggregation mechanisms reduces fibril toxicity

The paper under review studies how the naturally occurring polyphenol curcumin modulates spontaneous HttEx1 fibrillogenesis in cell-free assays. The study focuses on the aggregation of two HttEx1 variants with 32 or 48 glutamines and investigates the effects of curcumin on fibril morphologies and structures. The authors present experimental evidence that curcumin slows down HttEx1 aggregation and alters the structure of amyloid fibrils. Also, they present evidence that curcumin treated HttEx1 fibrils show reduced toxicity in cell-based assays (shown for one of the tested variants). Finally, the structures of compound treated HttEx1 were analyzed by SAXS and ssNMR measurements, indicating changes in the "fuzzy coat" of amyloid fibrils rather than in the polyQ segment of the fibril core.

The paper is of high interest for the scientific community working on amyloid aggregation mechanisms and aggregation modulators and therefore should be published in Nature Communications. However, I have the following major concerns that need to be addressed experimentally before this study is suitable for publication.

Major concerns:

The mechanism of action of curcumin needs to be addressed in more detail. Several lines of experimental evidence indicate that primary and secondary nucleation events are critical for the spontaneous self-assembly of HttEx1 fragments with pathogenic polyQ tracts (e.g., Q44-HttEx1) into amyloid structures. The authors present experimental evidence that curcumin alters the aggregation kinetics of HttEx1 in ThT assays. However, it remains unclear why the treatment with curcumin extends the lag phase of HttEx1 polymerization. Does curcumin directly bind to soluble MBP-tagged Q32-HttEx1 or Q44-HttEx1 fusion proteins? This could be addressed by fluorescent polarization assays using the intrinsic fluorescence of

curcumin. Does curcumin directly target the oligomers/prefibrillar HttEx1 structures that are spontaneously formed during the lag phase in the amyloid polymerization reaction? An important question is to address whether curcumin influences primary nucleation, secondary nucleation, or both? This needs to be dissected in more detail with cell-free assays. For example, it would be important to know whether the addition of curcumin after 5-6 h also has an effect on HttEx1 polymerization and influences the morphology of end-stage amyloid fibrils.

The authors present evidence that compound treatment potentially changes the structure of the N- and/or C-terminal domains that are exposed on the surface of HttEx1 fibrils (the “fuzzy coat”). However, it remains unclear whether the compound binds directly to the exposed N17 or the PRD domain. This could be addressed using N- or C-terminally truncated HttEx1 fragments that both form amyloid structures in cell-free assays (see e.g., Mariscal et al., 2022 Nat Commun). Also, dot blot assays should be performed with epitope-specific anti-HttEx1 antibodies to assess whether curcumin binding to HttEx1 amyloid fibrils alters their surface. Finally, compound binding studies with HttEx1 peptide arrays could be performed to better map the binding of curcumin to the “fuzzy coat”.

The authors investigated the toxicity of compound treated and untreated HttEx1 fibrils using MTT assays. They state that HttEx1 fibril treatment influences cell viability. However, these initial observations need to be substantiated. MTT assays measure the metabolic activity of intracellular enzymes; MTT reduction may indicate cellular dysfunction and not necessarily reduced viability. Experiments with compound-treated and untreated HttEx1 fibrils should be performed in additional well-established cell-based assays, e.g., with the highly sensitive CellTiter-Glo and/or the CytoTox-Glo assays (commercially available).

Minor concerns:

Line 196: The unit “hours” differs within the manuscript between hours, h, hrs. Please unify.

Line 205: A significant decrease is not shown for the condition 5 μ M/48 h.

Line 563: Spelling of Hoechst.

Line 573: Check sentence.

Figure 2: (a) vs (b) > Why did you test different sub-stoichiometric ratios and why is the x-axis changed? Where are the data points? What is shown (a fitted curve)?

Figure 2: (a) vs (b) > Why did you test different protein concentrations? (c-f) > 61 μ m? Why is the curcumin concentration in μ M not ratio? Do black and white scale bars both represent 50 nm? How many fibrils did you measure? (m) > Red dashed line is missing.

Figure 4: Check labeling of subfigures.

Supplementary Figure 1: (d) > Understanding would be easier with same color code. Where are the data points?

Extended Methods: ThT assay > Which bar graph in Figure 2b?

Version 2:

Reviewer comments:

Reviewer #4

(Remarks to the Author)

Jain et al.

Inhibitor-based modulation of huntingtin aggregation mechanisms reduces fibril toxicity

The authors have thoroughly addressed most of my concerns. One important point still needs clarification. The revised manuscript states (page 10, first paragraph): “Taken together, we observe that the mature HttEx1 fibrils exhibit toxic effects on both neuronal cell types, and that this toxicity is modulated significantly by curcumin inhibitors, resulting in less toxic fibril species.” This statement, however, is not in good agreement with the results shown in Fig. 3d. The quantification of neuronal network integrity (LUHMES cells) indicates that addition of Curcumin-treated fibrils does not significantly improve the neurite area (Fig. 3d). This needs to be clearly stated in the manuscript. Also, it still remains unclear whether the measured changes of MTT reduction in HT22 cells are an indication of reduced metabolic activity or reduced cell viability. In the figure legend (Fig. 3), the authors state that a cell viability assay was performed, while in the main text they state that MTT assays measure metabolic activity (page 9, line 197). It may well be that cell viability gets decreased when HT22 cells are treated with amyloidogenic HttEx1 aggregates. To be conclusive, this needs to be confirmed with an independent toxicity assay. As the manuscript’s title states that inhibitor-based modulation of huntingtin aggregation reduces fibril toxicity, showing a convincing effect of Curcumin on fibril toxicity is prerequisite for the publication of this study.

Reviewer #5

(Remarks to the Author)

The authors describe work that shows that nucleation and growth of misfolded HttEx1 peptide fragments, into aggregates of fibers, is modulated by the polyphenol curcumin. In particular, fiber structure characterization by ssNMR, TEM and SAXS support the conclusion that HttEx1 aggregation with curcumin present results in the decrease in fiber width in a curcumin concentration dependent manner.

The TEM study with micrographs and analysis displayed in figure 2 (for both Q44-HttEx1 and Q32-HttEx1 fibers)

consistently show a shift in fiber width distribution towards smaller values in the presence of curcumin with average width ranging from $\approx 10\text{-}14$ nm to $\approx 2\text{-}4$ nm.

The SAXS data appears consistent with the TEM data. However, I strongly recommend that the authors expand the discussion of the SAXS results and, in particular, in the context of three features as I detail below:

The SAXS data ($I(q)$ versus q) of HttEx1 aggregates in the absence and presence of curcumin (figure 2n) is displayed on a log-log plot. Three distinct regions can be identified.

First, at low- q , the SAXS intensity exhibits power-law behavior (although over less than a decade) with $I(q) \approx 1/q$ indicative of a fractal of dimension 1 and consistent with fiber scattering. The main caution is that the power-law behavior is over a relatively narrow q range less than 1 decade (and even a shorter range for the fibers in the presence of curcumin). The authors should point out that the evidence of fiber structure up to much larger length scales is coming entirely from the TEM images. (the SAXS data at very low q is consistent with a linear fiber structure for length scales between $\approx \text{constant}/q_{\text{min}}$ to $\approx \text{constant}/q^*$, where q^* is the beginning of the so-called Porod regime where the slope decreases abruptly from -1 to $\approx -3\text{-}4$. The constant is between 1 and 2×3.14 depending on the model used.)

Second, the SAXS data clearly shows a correlation peak somewhere around $q \approx 0.3 \text{ nm}^{-1}$ (the lack of tick marks on the q -axis log plot makes it hard to see exactly where the peak is in q -space). The analysis of the scattering (in the supplement) via a model that incorporates both the form factor of rods and the structure factor measuring fiber-fiber correlations seems reasonable. The analysis gives an accurate measurement of the distance between fibers. However, the authors should include a brief discussion on the fact that the very large width of the correlation peak implies that there may only be near-neighbor correlations between fibers (this could either be coming from bundles with very few fibers or from bundles with many fibers but with defects such as kinks/breaks along the fiber length). The TEM images are a better method of determining the width of the bundles.

Third, the authors should point out that the SAXS intensity versus q exhibits a clear high q regime after the correlation peak (i.e. small length scales of order the width of the fiber and smaller) showing an abrupt decrease in slope from -1 to somewhere between -3 or -4 (i.e. the transitions from the linear fiber to the Porod regime). The rod form factor incorporated in the SAXS analysis accounts for this rapidly changing slope regime and gives an accurate measurement of the radius of the rods without curcumin and with curcumin (figure 2o).

Finally, another important issue which is straightforward to take care of: I suggest that the authors add tick marks (along the both the x and y axes) so that any knowledgeable SAXS person will be able to immediately deduce the slope of the scattering data, characteristic peak position related to inter-fiber distance, and an idea of the correlation length from the width of the peak (even before getting the parameters from a detailed modeling process involving the form factor and structure factor).

Version 3:

Reviewer comments:

Reviewer #4

(Remarks to the Author)

Jain et al.

Inhibitor-based modulation of huntingtin aggregation mechanisms mitigates fibril-induced cellular stress

The authors have now comprehensively addressed all my concerns. The paper over time has improved significantly and is now ready for publication in Nature Communications. This study provides important mechanistic insights into the HttEx1 aggregation process as well as indicates that targeting of HttEx1 aggregation with the small molecule curcumin improves important molecular processes in cells.

Minor point:

The authors make the following statement in the text: "...there was a higher number of cell bodies positively stained with Calcein AM, which could reflect an improved viability, when compared to the group treated with fibrils formed in the absence of curcumin."

However, it would be nice to show the results from the Calcein AM stainings in the Suppl. Figures to support this statement.

Reviewer #5

(Remarks to the Author)

The authors have addressed all of my concerns. I recommend acceptance of their paper.

Response to reviewer comments

Below we provide a point-by-point response to the reviewer comments, which we reproduce verbatim.

Reviewer #1

In this contribution, van der Wel et al investigate the influence of Curcumin on the aggregation of the 32-residue long PolyQ containing construct Q32-Htt.Ex1 in terms of aggregation kinetics and structure, dynamics and cytotoxicity of the resulting fibrils. The addition of curcumin inhibits the aggregation, with increasing amounts of curcumin result in an extended lag phase. Likewise, the average diameter is reduced for fibrils grown in the presence of curcumin. On the other hand, the conformation of the individual glutamine residues within the fibrils as probed by solid-state NMR-spectroscopy is not altered by the addition of curcumin. However, Proline signals from the neighboring, mostly flexible proline rich domain seem to be slightly more enhanced in dipolar-based spectra of curcumin-grown fibrils, pointing towards slightly reduced mobility of this domain. Finally, cytotoxicity towards a neuronal cell culture of Q32-Htt.Ex1 fibrils grown in presence of curcumin is reduced with respect to that on fibrils grown in the absence of curcumin, and most of the fibrils regardless whether grown in presence or absence of curcumin do not enter the cells. These reduced dimensionality of curcumin-grown fibrils is explained by the hypothesis that addition of curcumin favors the formation of beta-hairpins, which slows down the aggregation of Q32-Htt.Ex1. This article presents a sound study of the potential effects of a "fibrillation inhibitor" on the aggregation process of Q32-Htt.Ex1 in vitro. The authors do state very clearly that they have studied in vitro effects, that aggregation is only slowed down and not prevented, and that the observed reduced toxicity of in vitro grown fibrils can not directly transferred to claims on in vivo effects. Most results are well justified and hypotheses are clearly indicated as such.

We appreciate these very positive comments from the reviewer.

Only the conclusion that the Pro-rich domain is less flexible in fibrils grown in presence of curcumin should be handled with more care: at least in the second batch of fibrils this effect seems to be negligible. And in any case, a quantification (relative intensity of the integral of the Pro peak with respect to one or more significant Gln peak(s) would be helpful in this respect).

We have performed the suggested peak integration and added it to the revised manuscript as Supplementary Figure 7 in the SI. We have also expanded and revised the discussion of these results (p. 12), citing the new figure and more clearly mentioning the variability of the effect.

On a minor point: Figure legend 3a : to my understanding, Figure 3a shows only results on fibrils grown in absence of curcumin, so the second half of the sentence (and HttEx1 fibrils prepared in presence...) should be omitted.

We thank the reviewer for noticing this mistake, which we have now corrected.

Reviewer #2

The basic flaw in this work is the outdated references to Huntington's disease, that are repeatedly either >25 years old or just wrong.

This statement by the reviewer is not an accurate description of our manuscript. References from over 25 years ago would be from 1998 or older. Of the 65+ citations in our original paper, only one was that old – our citation of Anfinsen's classic work from 1973. The next-oldest reference was from 1999, with the rest being from the 2000s, including many from the last few years.

That being said, to accommodate new references in the revised manuscript, we have gone through the cited works and removed some less important and older entries (including the reference to Anfinsen's paper).

*"Huntington's disease (HD) is a neurodegenerative disorder in which a mutated fragment (exon 1) of the huntingtin (HttEx1) protein undergoes misfolding and aggregation."
No, HD is not caused by protein misfolding and aggregation.*

The reviewer seems to interpret the cited sentence as claiming a causal relationship between protein aggregation and HD, however this was neither intended nor is it actually written as such.

It merely notes that the aggregation of mutant protein into inclusions (= aggregates) is generally considered a hallmark of HD, both historically and at this moment. These inclusions are assembled from fragments of the mutant huntingtin protein, in which the proteins are in a structure that is unlike their nature fold (i.e., 'misfolded'). As such, the quoted sentence seems to us correct and not 'out of date'.

For the benefit of readers, we have added several additional citations to the introduction for review articles on the topic of Huntington's disease and its mechanisms. These include references 4 and 5, which are recent reviews that discuss protein aggregates as a key feature of HD, while also examining the complexities of understanding their role in the disease:

Tabrizi, S. J., Flower, M. D., Ross, C. A. & Wild, E. J. Huntington disease: new insights into molecular pathogenesis and therapeutic opportunities. *Nat Rev Neurol* **16**, 529–546 (2020).

Bates, G. P. *et al.* Huntington disease. *Nature Reviews Disease Primers* **1**, 15005 (2015).

Two other possible questions are whether the misfolded state of the protein is considered relevant to the disease, and whether it may potentially even be a causative agent.

In terms of the relevance of the aggregates for disease:

- They are seen as essential biomarkers of disease progression. We make this claim based on the fact that leading researchers and HD foundations are investing in the costly development of PET ligands that bind these aggregates, in the aim of tracking the effectiveness of protein lowering strategies as a HD treatment strategy. (see newly cited reference 7)
- Other groups are developing (as biomarkers) ‘amplification’ or ‘seeding’ measurements that aim to detect the seeding-capable protein inclusions in CSF (newly cited references 8-9)

We have added additional comments and citations to the introduction of the paper, to better connect our work to other cutting-edge HD research.

The question of aggregates as causative agents is addressed below.

1. There is no correlation between aggregate presence and neuronal dysfunction in HD brains.

We are not aware of published studies that thoroughly and systematically correlate the aggregate burden to neuronal dysfunction in Huntington disease patient brains. A huge challenge here is the required technology that could (1) measure the aggregate burden and (2) measure neuronal dysfunction within (living) patients. We ourselves are involved (at the periphery) in the ongoing effort to develop the technology (PET imaging ligands) that can help provide this level of insight, so we are particularly aware of the challenges in making such analyses possible.

2. There is evidence from 2006 on that any aggregates may be protective in neurons.

As already mentioned in response to the previous point – the detection of aggregates is not an easy task. The classic HD literature (from around 2006) often relied on the detection of inclusions by using fluorescently-tagged proteins in cultured cells. Aside from concerns about the possible effect of the fluorescent tags, a big issue is that aggregates below the resolution limit of classic (confocal) microscopy were simply invisible. Here, the term “any aggregates” (as used by the reviewer) is worth a comment: in confocal microscopy studies one type of aggregates (large inclusion bodies) may be visible, but smaller aggregates would not be. This is not just conceptually to be expected, but was actually demonstrated in elegant work by groups using modern super-resolution techniques not available in the earlier days of HD research. Perhaps most interesting in this context is the work by W.E. Moerner and colleagues, which became possible after 2010:

Duim, W. C., Jiang, Y., Shen, K., Frydman, J. & Moerner, W. E. Super-resolution fluorescence of huntingtin reveals growth of globular species into short fibers and coexistence of distinct aggregates. *ACS Chem Biol* **9**, 2767–2778 (2014).

Sahl, S. J., Weiss, L. E., Duim, W. C., Frydman, J. & Moerner, W. E. Cellular Inclusion Bodies of Mutant Huntingtin Exon 1 Obscure Small Fibrillar Aggregate Species. *Sci. Rep.* **2**, (2012). [now cited as ref. 16]

New insights enabled by these techniques (as well as modern in-cell EM as well as amyloid-seeding assays) allow the detection of small aggregates that would have been invisible in cellular studies from the early 2000s. For example, the most well-known and widely-cited study on this topic is from 2004 by the Finkbeiner group (new ref. 13), who published on the apparent protective effect of inclusion bodies in neuronal cell studies. That study, like others in the literature, is likely handicapped by the inability to detect isolated fibrils, focusing only on large inclusions.

A key point here is therefore: when one considers “any aggregates” then one needs to deploy techniques that can detect (and distinguish?) different types of protein aggregates (from large micrometer-sized inclusion bodies, down to individual fibrils or prefibrillar aggregates).

We have added extra information to the introduction, explaining some of the above considerations and including more citations to relevant recent HD literature (see page 3).

3. In human iPSC derived neurons, and human derived model lines, there are clear HD phenotypes, and no protein inclusions.

Also here, a specific citation might have been useful to fully understand the origin of this comment. A review article by Steven Finkbeiner from 2022 concluded that there is mixed evidence of inclusion body formation in patient-derived (iPSC) neurons (Fac Rev. 2022; 11: 16). Reportedly, some studies could not detect aggregates using antibody assays, whereas others could. It is suggested (by Finkbeiner and co-authors) “... *that aging and the brain microenvironment may be required for pathological mHTT IB formation.*” In other words, this review article from 2022, co-written by one of the authors of a key paper on the topic of protective inclusion-body formation, described mHTT IB formation as pathological.

We have not made a specific change to our paper as we consider this topic beyond the scope of the current work, with an eye on avoiding an excessively detailed introduction section.

4. in 2016 Human GWAS studies, now at over 16,000 patients, there is no evidence that any proteostasis pathways are modifiers of disease onset, nor severity.

The main signals seen in GWAS studies so far have been traced to proteins (genes) that modulate somatic expansion of the mutant HTT gene. In part based on such findings, there is now a growing appreciation for the role played by somatic expansion in patients: this process seems to generate even-longer gene expansions than inherited. An as-yet underdiscussed implication of this finding (in our view) is that this does not change the fundamental question of **how these expanded genes lead to disease**. A potentially interesting finding (also noted in the already-mentioned and newly cited review by Tabrizi, Flower, Ross and Wild; ref. 4) is that at least one study noted a **reduced aggregation** being associated with the knockout of a GWAS-identified modifier of somatic expansion (MSH3). Thus, the presence of

an important **role for genetic instability does not preclude a role for aggregation of the resulting mutant proteins** in the disease.

Aside from citing the mentioned review article, we have not made a specific comment in the paper, to avoid an excessively detailed introduction section.

5. There was one transgenic model of huntingtin Exon1 from 1996, for which no control strain exists. Unfortunately, this has become dogma because it follows an simplistic amyloid hypothesis for which there has been a plethora of contradictory evidence in the last 15 years.

First, the reviewer here seems to describe the possible role of exon 1 as ‘dogma’, which seems contradictory to the reviewer’s claim that the HD field has moved on from this idea. As already noted, in our view, the focus on HTT exon 1 and related fragments is quite alive in the field. This is apparent in modern review articles of HD pathogenesis; notably including those written by key researchers working on other aspects of the disease (e.g. the already-mentioned reviews by Finkbeiner and colleagues, and Tabrizi and colleagues).

Second, the relevance of the number of exon-1 mouse models is unclear to us. There have been many different variations on HD mouse models (e.g. reviewed in DOI 10.1038/nrn3570), featuring not only full-length HTT but also various types of N-terminal fragments. Unless we misunderstand the reviewer’s comment, it does not seem quite accurate to suggest that there is only one HD mouse model based on huntingtin exon 1, given that the (admittedly related) R6/1 and R6/2 constructs (with different polyQ lengths) exist. What the reviewer means with a lack of ‘control strain’ is also not clear to us, or how this is relevant to our manuscript.

Third, the reviewer mentions the ‘simplistic’ amyloid hypothesis as being contradicted by a plethora of evidence. We presume that this comment at least in part refers to the status of this famous hypothesis also in other ageing disorders. In AD the amyloid hypothesis may be controversial for some, but it is hardly considered disproven, given the recent approval of multiple new antibody-based treatments that are thoroughly grounded in this hypothesis. Thus, beyond HD, the amyloid hypothesis seems to be alive and not disproven ‘in the last 15 years’.

6. In symptomatic patient brains who died prematurely, there is little evidence of protein aggregate pathology.

We are unsure of the exact studies the reviewer has in mind when providing this comment, as studies of prematurely deceased HD patients are rare but very interesting. We have revisited some of the pathology-related literature for HD. A fairly recent review article focused on HD patient pathology (DOI :10.1111/bpa.12426) from 2016 seems to stress the relevance of protein inclusions (or aggregates). We cannot find any direct support for the claim that there is good evidence for a **lack** of aggregates. Here, we also reiterate the point raised above, i.e. the need for tools suitable to detect all types of aggregates that may be relevant (i.e. not only large inclusions that may form at late stages of disease, potentially as a rescue mechanism).

In summary:

Having done our best to address the numbered concerns raised by the reviewer, we would like to add a more general response. While we are familiar with some of the presumed references that this reviewer is considering, it would have been useful to have specific sources for the above claims.

The past decade has seen tremendous progress in the development of molecular-level techniques that look beyond the classic assays used to detect protein inclusions in clinical and cell biological research. The classic views in these fields are based on the detection of micrometer-sized inclusions detected via immunostaining (DiFiglia 1997; ref 6) or confocal microscopy. Careful biophysical studies on many types of proteins (including HTT fragments) have shown that these larger inclusions contain many individual aggregates that are just nanometers wide. In isolation, such filaments are invisible in those assays. Any assay that relies on classic methods to detect inclusions would miss all but the larger clusters of protein aggregates. (Notably, experts in protein amyloid formation typically find that smaller aggregates are more harmful in terms of their cellular uptake, cell-to-cell transfer, and apparent cytotoxic activity).

The second problem is curcumin which has been funded and studied by the NIH for a decade or more and proven not to be bioavailable. I will politely refer to curcumin research in general as “controversial”.

We very much appreciate the effort by the reviewer to be polite in their phrasing of this comment. In terms of a response to the scientific context: reviewer 1 specifically noted: *“The authors do state very clearly that they have studied in vitro effects...”* This comment by reviewer 1 describes the fact that our aim is to use this polyphenol as a tool toward understanding mechanistic and molecular principles. Our aim is **not** to promote the use of curcumin for any particular clinical purpose (e.g. the well-known challenges related to its limited bioavailability).

In response to the misunderstanding by reviewer 2, we have re-checked the manuscript for any potentially suggestive comments otherwise and corrected these where applicable. (E.g. slightly rephrasing the abstract, removing a mention of ‘candidate amyloid inhibitors’ from the Discussion on p. 13, etc). Additionally, we have added the following statement to page 21, along with a citation of a review article that discusses the limitations (and uses) of curcumin:

“That said, the focus of the current work is on the use of curcumin as a model compound to evaluate possible (unexpected) side-effects of inhibition on the misfolding process, rather than advocating the actual use of curcumin as a direct treatment of HD⁴⁰.”

REF 40: Anand, P., Kunnumakkara, A. B., Newman, R. A. & Aggarwal, B. B. Bioavailability of Curcumin: Problems and Promises. *Mol. Pharmaceutics* **4**, 807–818 (2007).

We also cite this report in the introduction, at the first mention of curcumin (p. 5), along with the added comment: *“... its limited bioavailability can present a challenge in clinical applications [ref].”*

We anticipate that these edits reduce the chance that readers misinterpret our aims and intentions in this work.

Figure 2: there is no consideration of unbiased quantification, 2b is a single measure, somehow presented as a bar graph.

Figure 2b was used to visualize the half-time values of aggregation, a parameter derived from the data in panel (a) of the same figure. In the revised text we decided to omit panel (b) as the information was already visible in panel (a).

Figure 3: the use of Q32 exn1 model is strange, because this is not the pathogenic threshold of CAG allele expansion seen clinically at 37 or more, with a mean of 43. This just proves that massive overexpression of a synthetic fragment will cause an aggregation event that is irrelevant to human disease. No control of <Q21 wild type allele length.

It is well known in the literature that the HttEx1 protein has a potent tendency to aggregate (in vitro) that depends on the polyQ segment length, with very short polyQ lengths resulting in soluble proteins. However, an interesting (but known) feature of these proteins is that they aggregate quite well for polyQ segment lengths shorter than the disease threshold, but longer than typical wild-type (around 20 glutamines). This feature is leveraged quite frequently in 'in vitro' studies, to facilitate in-depth studies of molecular mechanisms. In response to comments by this and reviewer 3, we have more prominently featured our data on Q44-HttEx1 in the paper (where available). See also our comments to reviewer 3, below.

3d: bar graphs with lower error bars removed for aesthetic reasons, not appropriate. Same figure T-test analysis on three conditions, which should be ANOVA. Then a complete different format appears for 3F. More care should be taken in figure and data presentation.

We have checked Figure 3 and updated it to have a more uniform appearance and to conform to the Nature Comms reporting guidelines. An ANOVA test was indeed performed for the analysis (noted in caption).

No mention of effects on exon1 PTMs, which are known to affect aggregation.

We have studied the effect of PTMs in prior work, but the reviewer is correct to note that this was not a focal point of the current study. To avoid excessive speculation, we would prefer to not comment on this topic in absence of relevant new experimental data.

Reviewer #3

In this well-written article, Jain et al. provide compelling evidence that curcumin, a known inhibitor of amyloid formation, alters the mechanism of aggregation by huntingtin protein in vitro. In the presence of substoichiometric concentrations of curcumin, amyloids formed with delayed kinetics, an altered structure, and reduced toxicity. These findings will stimulate

further work into the mechanism of potentially therapeutic small molecule-based modulation of amyloid formation.

We appreciate the positive comments from the reviewer, and in particular the recognition of our focus on mechanistic insights, as well as the potential benefits of such work for future therapeutic research.

Critiques:

1. In the abstract and introduction, it is stated that deposits/deposition/inclusions are toxic, while the authors elsewhere intend to be agnostic about the nature of toxic species. I would therefore replace these terms with something less specific, such as "aggregates" or "misfolded species".

We have revisited the abstract (and introduction) and made changes as suggested, regarding the precise terminology. In response to reviewers 2 and 3 we have also provided more (literature) context where necessary.

2. Structural features of amyloids, such as the specific polymorph produced, tend to be extremely sensitive to concentration, temperature and other conditions of the amyloid assembly reaction. It is therefore desirable to provide some explanation of the conditions used for assembly, and rationale for choosing them, in the first results section.

Like other amyloidogenic proteins, it is indeed the case that HttEx1 polymorphs depend on the temperature and concentration. Here, we were informed by prior published conditions (from our 2020 paper in JMB; ref 19) and aimed to use constant conditions where a uniform polymorph is formed. We have now added a brief explanation of the condition to the results section (page 5-6), along with citations to relevant prior work on HttEx1 polymorphism.

" From previous studies ^{11,12,14,19}, we know that HttEx1 aggregation can produce different polymorphs, with notable effects of the concentration and temperature¹⁹. Here, we employed a concentration (61 mM) in PBS (pH 7.4) and (room) temperature where we reproducibly can form a dominant type of polymorph¹⁹. " [page 6]

3. The increased lag phase in the presence of curcumin is interpreted as evidence that curcumin inhibits primary nucleation. However, a qualitative analysis of ThT kinetics does not suffice to make this conclusion. As exhaustively explained by Linse, Knowles and colleagues (see e.g. <https://doi.org/10.1039/C4CP05563B>), the lag phase is a product of primary and secondary processes; it is not a simple indicator of primary nucleation rates. The curves shown here can, in principle, be fit to kinetic equations (e.g. AmyloFit) to distinguish the effects of curcumin on impacts primary or secondary processes.

We thank the reviewer for this suggestion. We have indeed been performing analyses with AmyloFit for our aggregation reactions (e.g. Boatz et al 2020 JMB), which indeed suggests a prominent role for secondary nucleation. We have now mention the above, discuss the role of secondary nucleation and have added results from such an AmyloFit analysis of the uninhibited reaction in Supplementary Figure 1a-b. The analysis of the curcumin-inhibited reaction is more complex due to the presence of the fluorescent inhibitor (see also Supplementary Fig. 1) and a lack of

concentration-dependent data that are needed for comprehensive modelling of the aggregation mechanism.

As such, we have modified our manuscript such that we mention the role of secondary nucleation and no longer claim that (only) primary nucleation is affected by the inhibitor. We also have gone through the manuscript and removed or edited any discussion that overly focused on primary nucleation in particular (e.g. page 13 of the Discussion).

4. I am dismayed that the authors focus on Q32 Htt throughout this work, while relegating data for Q44 Htt to the supplement. They do not provide a justification for this choice, even though the latter is uniquely relevant to disease. Likewise, in the discussion, they choose to interpret Q32 with respect to their prior findings with Q44, while oddly avoiding any mention of the present data on Q44.

We chose to work with Q32-HttEx1 for the structural and toxicity studies based on the need for decent sample sizes for both experiments, with the ssNMR also requiring isotopically labeled protein. This was more challenging to achieve with Q44-HttEx1 as it gives us a lower yield in *E. coli* and is less stable (even as a fusion protein). As a consequence we had opted to focus on the Q32-HttEx1 data, where we had a more complete dataset. We have now partly revised this strategy, by putting more of the Q44-HttEx1 data in the main text and discussing the relevant findings also in more detail. We thank the reviewer for encouraging this approach, and also for suggesting interesting mechanistic points here and below.

5. Although the assay used to analyze toxicity is not atypical for the amyloid field, do we have any reason to believe that it assesses a disease-relevant mechanism? I.e. is there evidence that extracellular Htt aggregates contribute in any way in vivo? Is there known to be a correlation between extracellular and intracellular toxicity? If not, please clarify the relevance of results using this toxicity assay, even if just to further indicate that the fibrils are different.

We have slightly expanded our discussion of the type of toxicity assay and how it may relate to the disease condition (page 20). As suggested by the reviewer, our main purpose with this assay is to identify differences in the biological properties of the fibril polymorphs, rather than necessarily mimic the HD disease conditions. That said, there may actually be relevance from a disease point of view. Although the protein aggregation in HD is commonly described as intracellular, there is a growing interest in the ability of these aggregates to end up outside neuronal cells and/or travel between cells. In terms of the former, aggregates in CSF are the focus of assays being developed as biomarkers for disease progression (E.g. work by the Wanker group in Berlin). Secondly, there is a growing interest in the idea that disease progression may follow a prion-like mechanism, in which aggregates travel to neighboring cells (Alpaugh et al., 2022; ref 66). Moreover, HD brain cells degenerate and undergo cell death, resulting in (intracellular) aggregates being released and being detected in the extracellular matrix (reviewed by Tabrizi et al., 2020; ref 4). In summary, the study of extracellular aggregates in HD (and their role in disease) is an active field of investigation (although perhaps still considered controversial by some). These topics (and references) have been added to the manuscript on page 20.

6. Do the stated concentrations of fibrils used in the toxicity assay refer to the concentrations of resuspended fibrils as determined experimentally, or do they simply assume 100% conversion of starting protein? Please clarify if (and how) the concentration of fibrils was normalized to ensure that equal amounts of treated vs nontreated fibrils were compared. This would help in interpreting the reduced toxicity of curcumin-treated fibrils with respect to structure rather than a possible reduction in the extent of fibrillization. I do find that the subsequent quantitation of fibril binding/uptake by flow cytometry goes a long way to address this concern.

The concentration of the fibrils used in toxicity assay refers to the concentrations of the resuspended fibrils, which were indeed based directly on the concentration of the monomeric protein. Indeed, we assume 100% conversion of the starting protein. The concentration of the fibrils was not normalized except to use identical amounts of protein in each experiment. The concentration of the monomeric protein with and without curcumin was determined before the aggregation. After the aggregation process, the fibrils were pelleted by centrifugation. To look for unaggregated monomers, we performed both SDS-PAGE and LC-MS on the supernatant obtained from pelleting the fibrils. In both cases, we were unable to detect any unaggregated HttEx1 monomers, supporting the assumption that essentially 100% converted into fibrils.

This observation is also consistent with prior studies on polyQ aggregation that reported the critical concentration of monomers at equilibrium with the fibrils (C_r). For instance, Landrum & Wetzel (2014; DOI 10.1074/jbc.C114.552943) report the C_r for the polyQ peptides of different polyQ segment lengths, with values in the range of 0.3 to 0.8 μ M for polyQ31 and polyQ42 peptides. Although these model peptides are distinct in their aggregation behavior, we think it reasonable to expect similar sub- μ M values for our HttEx1 protein fibrils with similar polyQ lengths.

7. The fibril widths extracted from TEM data are assumed to reflect the widths of the amyloid cores. This assumption is inconsistent with published work showing that TEM over-estimates the widths of fibril cores (e.g. <https://doi.org/10.1038/s41593-023-01341-4>, <https://doi.org/10.1002/1873-3468.13675>). This is importance for the structural interpretations that rely on these measurements.

This specific question has been a topic of investigation and discussion in our group since we started building models for the protein fibrils (as published in 2016 and 2017). How do we interpret negative stained images where we detect the stain, rather than the protein? Ultimately, we concluded that the balance of evidence is that the stain must penetrate between the flanking domains, while being unable to enter the tightly packed amyloid core. This is the rationale for interpreting the TEM widths as amyloid core widths. The above conclusion was based on a number of considerations (reported in our prior work from 2016, 2017 and 2020):

- SSNMR shows clearly that the flanking domains of HttEx1 fibrils are very flexible and dynamic (at least for thinner polymorphs, with widths below 10 nm), implying that they cannot be tightly packed on the fibril surface. This is

also supported by ssNMR measurements that probe solvent exposure, and which showed the flanking domains to be highly solvent exposed.

- Conversely, ssNMR showed that the polyQ core itself is extremely rigid (akin to crystal lattices) and inaccessible for solvent interactions.
- The main flanking segment that does experience some degree of potential ordering is the C-terminal flanking domain. SSNMR studies of fibrils show this proline-rich domain to adopt an (extended) polyproline II structure, which would in itself be multiple nm long (being formed by >20 proline residues). So, if these segments were contributing to the TEM-detected core width, then this would not leave any width attributed to the polyQ itself.

We also have useful data from other orthogonal methods:

- We have performed SAXS data on fibrils with and without curcumin inhibition, and added these results as new data to this paper. The new SAXS data are shown in Figure 2n-o and discussed also in the SI (tables and methods). The obtained data support the conclusions drawn from TEM analysis, in terms of the fibril dimensions as well as the impacts of curcumin.
- In the current and published work, we have reported fibril widths of 5-12 nm from negative stain TEM for the fibril cores of Q44-HttEx1 fibrils. A recent cryo-EM study by the group of Lashuel has reported the widths for their Q43-HttEx1 fibril cores as varying between 5 and 13.5 nm (DOI 10.1021/jacs.2c00509; ref 15). Although different preparations, the observed dimensions and architectures seem very consistent, and do not give much reason to suspect significant overestimations of the fibril core width.
- In preliminary cryo-EM measurements (unpublished) we find data similar to the Lashuel paper, indicating a good match between the TEM-widths and the ordered polyQ core of the fibrils. These data are too preliminary to include at this time.

We greatly appreciate the fact that this reviewer provides specific references with their feedback and questions. The reviewer points out two interesting papers on other amyloid fibrils. One is focused on the TDP-43 fibrils (Kumar et al 2023; cited as ref 57), where stripping of flanking domains reveals indeed a thinner fibril core than at first apparent from negative stain TEM. A notable feature of this paper is that it discussed the observation that the amyloid core of the TDP-43 fibrils is so buried by its flanking domains (or 'fuzzy coat') that it is reportedly not detected in ThT assays, unless the fuzzy coat is removed by protease treatment. To quote the authors: "*heavily decorated by the structured domain of the protein (NTD and RRM)s, which mask the amyloid core of the filaments, thus rendering the amyloid core inaccessible to amyloid-specific dyes such as ThS/ThT and CR* [p. 984]." The folded nature and dense packing of the flanking domains in TDP-43 fibrils make the latter quite different from HttEx1 fibrils (see above), and explain the difference in the ability of stain to reach the amyloid core.

The second cited paper by Al-Hilaly and co-workers looks at fibrils formed by a fragment of the tau protein. Here, it is difficult for us to comment in much detail, except to note that the architecture of tau fibrils (e.g. as seen by modern cryo-EM) is rather distinct from the structure of HttEx1/polyQ fibrils (see e.g. our latest perspective on this in a pre-print on bioRxiv; Helabad 2023; <https://doi.org/10.1101/2023.07.21.549993>).

Altogether, we are inclined to retain our primary assumption that negative stain analysis yields widths that mostly represent the amyloid core size. We have added additional discussion of the above considerations and context to the manuscript (page 4). In the Discussion we also expanded our discussion of the TEM width interpretation with more explanations, more citations and additional caveats (top of p. 16).

8. The interpretation of Q32 fibrils as lacking a hairpin is inconsistent with multiple prior studies showing that polyQ segments down to 26 residues in length aggregate via a β -hairpin nucleus. This was previously reviewed by Matlahov and Van der Wel 2019 (authors on the present paper). I do not consider fibril widths obtained from TEM sufficient to overturn these prior studies. If one accounts for the likelihood that the stain does not fully penetrate to the core, then the core will be smaller than the observed width, and is not inconsistent with a turn-containing model.

An important detail of the mentioned prior work in which β -hairpin formation was analyzed (finding the 26-residue switchover point) is that those were studies performed with polyQ peptides that lacked the complete context of the HttEx1 protein used in this work (and which is relevant to HD). It is well established that the HttEx1 flanking segments have dramatic effects on the mechanism and kinetics of aggregation, with the N-terminal segment enhancing aggregation and the C-terminal segment inhibiting it. The latter effect appears to be mediated at least in part by a change in the structural ensemble of the preceding polyQ segment. As such, the present and past data are not inconsistent, given that one cannot directly compare model polyQ peptides and 'proper' HttEx1. (We have added a comment about the difference between model polyQ peptides and polyQ in HttEx1 context on page 18 of the revised manuscript)

Regarding specifically proving the presence or absence of β -hairpins: We have previously published the use of specialized (and time-consuming) ssNMR experiments that allowed us to determine the presence of β -hairpin structures in Q44-HttEx1 fibrils (Hoop et al 2016. DOI 10.1073/pnas.1521933113 ; ref 26). In ongoing and unpublished work we apply such experiments to Q32-HttEx1 fibrils, which do indicate a structural change relative to the Q44-HttEx1 fibril core that may be due to a change from β -hairpin structures. However, we prefer to delay publication of such data until we have the full dataset that allows for a thorough and quantitative analysis.

9. Likewise, the explanation for why Q32 may disfavor a hairpin seems inconsistent with prior in vitro and in silico observations that polyQ in this length regime is a highly collapsed globule (e.g. <https://doi.org/10.1073/pnas.0608175103>, <https://doi.org/10.1073/pnas.0900678106>,

<https://doi.org/10.1016/j.jmb.2009.08.034>) driven by intramolecular H-bonding. It is therefore not necessarily true that a hairpin would be entropically disfavored relative to an extended strand (I would argue the opposite). In any case, it will depend on whether the extended strand can be stabilized intermolecularly and therefore depend on concentration.

The reviewer here raises another very interesting point that is a topic of discussion in the polyQ biophysics field. The idea that β -hairpins play a significant role in polyQ aggregation goes back to the early papers by Perutz in the 1990s, who on basic principles suggested the idea. As such, there has been considerable effort to detect (by experiments) evidence for β -strand or β -turn conformations in the collapsed globular state that characterizes non-aggregated polyQ (and polyQ proteins). Nonetheless, to the best of our knowledge, no such evidence has been unearthed as yet. (This was for instance a topic of interest in our prior work with Ronald Wetzel, where solution-state CD and liquid-state NMR were used to probe the ensemble of structures in soluble polyQ. Even in presence of β -hairpin-stabilizing mutations no turns or other beta structure was detected. See Fig. 6 in 10.1016/j.jmb.2016.12.010)

As such, in our view, there is good evidence for the formation of a collapsed globular state (sometimes described as a 'tadpole' like structure), but no (experimental) evidence that a β -strand or β -hairpin structure is populated to any significant degree within such globules.

We have carefully read the cited papers (DOI codes from the reviewer), but the experimental evidence for **intramolecular hydrogen bonding** in the provided references is not immediately obvious to us. That said, recent solution NMR studies have provided some interesting insight into the structure and hydrogen bonding of polyQ chains (e.g. DOI 10.1038/s41467-019-09923-2). These studies have reported on intramolecular hydrogen bonding, but in those cases these were mostly interactions between sidechains and backbones; very different from β -strand or β -hairpin formation.

Given the complexity of this issue, we opted to remove some statements about the mechanistic aspects behind β -hairpin formation, which may be considered speculative or controversial. A very in-depth discussion (with appropriate citations) goes beyond the scope of this (already long) manuscript.

10. Though not discussed, the median fibril width for Q44 treated with curcumin is between 3 and 4 nm as shown in Supp Fig 2. This width suggests more than one turn and/or arc per monomer, consistent with the nucleus model of Kandola et al. 2023 (<https://doi.org/10.7554/eLife.86939.1>) for polyQ >36. The effect of curcumin could therefore be interpreted as inhibiting the transition from this short-stranded core to the longer single hairpin core of mature polyQ amyloid. Stated differently, the data appear to be consistent with curcumin stabilizing an arch-containing polymorph that is otherwise only transient. Assuming this species grows slower than the mature amyloid polymorph, this explanation is also consistent with the slower kinetics in the presence of curcumin.

As suggested by this reviewer, we have added more discussion of the Q44-HttEx1 results, comparing it to the Q32-HttEx1 case (moving some of the data to the main text figures as well). On page 19, we now briefly discuss that for Q44-HttEx1 treated

with curcumin, there is a possibility that the core may contain multiple turns or arcs, based on the width analysis from the TEM. As noted above, a more detailed analysis of the structure is ongoing, aided also by an integration with atomistic modelling which we recently posted to bioRxiv (see above; together with the group of Prof Markus Miettinen in Bergen, Norway).

We also added an expanded discussion of the idea that arc or arch structures may play a role in the mechanism, representing a pathway that may be enhanced by curcumin. (page 18-19)

In conclusion, we want to thank the reviewers for their constructive comments and feedback, which inspired us to revisit our data, interpretations and presentation/formulations in some detail. We hope that the revised manuscript is clearer in key points, much better positioned relative to prior work, and now acceptable for publication.

Response to reviewer comments

Below we provide a point-by-point response to the reviewer comments, which we reproduce verbatim.

Reviewer #1

The authors have addressed all my concerns and those of the other two reviewers. Thus, the manuscript is in my point of view ready for acceptance.

We appreciate the reviewer's positive comments about the paper and our prior revisions.

Reviewer #3

This revised version of Jain et al. adequately addresses my critiques of the original submission, and is appropriate for publication. I commend the authors on their thoughtful, professional, and very thorough handling of the reviewers' comments.

We appreciate the reviewer's positive comments about the paper and our prior revisions.

Reviewer #4

Jain et al.

Inhibitor-based modulation of huntingtin aggregation mechanisms reduces fibril toxicity

The paper under review studies how the naturally occurring polyphenol curcumin modulates spontaneous HttEx1 fibrillogenesis in cell-free assays. The study focuses on the aggregation of two HttEx1 variants with 32 or 48 glutamines and investigates the effects of curcumin on fibril morphologies and structures. The authors present experimental evidence that curcumin slows down HttEx1 aggregation and alters the structure of amyloid fibrils. Also, they present evidence that curcumin treated HttEx1 fibrils show reduced toxicity in cell-based assays (shown for one of the tested variants). Finally, the structures of compound treated HttEx1 were analyzed by SAXS and ssNMR measurements, indicating changes in the "fuzzy coat" of amyloid fibrils rather than in the polyQ segment of the fibril core.

The paper is of high interest for the scientific community working on amyloid aggregation mechanisms and aggregation modulators and therefore should be published in Nature Communications. However, I have the following major concerns that need to be addressed experimentally before this study is suitable for publication.

We appreciate the positive comments from the reviewer about the high interest of our findings for the community.

Major concerns:

The mechanism of action of curcumin needs to be addressed in more detail. Several lines of experimental evidence indicate that primary and secondary nucleation events are critical for the spontaneous self-assembly of HttEx1 fragments with pathogenic polyQ tracts (e.g., Q44-HttEx1) into amyloid structures. The authors present experimental evidence that curcumin alters the aggregation kinetics of HttEx1 in ThT assays. However, it remains unclear why the treatment with curcumin extends the lag phase of HttEx1 polymerization.

We agree with the reviewer that the mechanism of action of how curcumin works is important and interesting. A detailed molecular analysis of how or why a particular compound changes the aggregation behavior of amyloidogenic polypeptides is valuable but also non-trivial. In our view, the current paper presents a series of new findings that are notable and important even in absence (for now) of detailed knowledge of the precise mechanism of inhibition. Our current findings provide significant insights into the effect of curcumin on the HttEx1 aggregation process, toxicity, morphology, and structure. These findings have relevance not only to research into amyloid inhibitors, but also pave the way for further studies of the connection between fibril structure and cytotoxic mechanisms. Thus, our labs have ongoing research in which we probe the molecular mechanism of curcumin (and other inhibitors) as well as the structural features that determine fibril toxicity.

In a few places in the Introduction we slightly rephrased sentences to clarify the focus of our study and avoid a possible interpretation that we aim for a complete molecular understanding of the mechanism of action.

Does curcumin directly bind to soluble MBP-tagged Q32-HttEx1 or Q44-Httex1 fusion proteins? This could be addressed by fluorescent polarization assays using the intrinsic fluorescence of curcumin.

The reviewer raises an interesting point, and we added new data to the paper to comment on this. Compared to dissolved curcumin (in absence of protein), we do see a modest increase in fluorescence in presence of our non-aggregated MBP-HttEx1 fusion protein. This seems to suggest a (weak) interaction. However, the amount of fluorescence is still much lower than for that (same amount of) aggregated protein. We have added the fluorescence emission profile of curcumin in presence and absence of the fusion protein in the revised Supplementary Fig .1(g), which is mentioned on page 7 (top paragraph) of the main manuscript.

Does curcumin directly target the oligomers/prefibrillar HttEx1 structures that are spontaneously formed during the lag phase in the amyloid polymerization reaction? An important question is to address whether curcumin influences primary nucleation, secondary nucleation, or both? This needs to be dissected in more detail with cell-free assays. For

example, it would be important to know whether the addition of curcumin after 5-6 h also has an effect on Httex1 polymerization and influences the morphology of end-stage amyloid fibrils.

As already noted, we agree that a deeper mechanistic study of these (and other) inhibitors is important and timely. In ongoing research we are performing systematic studies of curcumin and other (potential) inhibitors. However, to do so thoroughly requires extensive and careful experiments (varying protein concentrations, inhibitor ratios, with and without seeding, and with (as suggested) varying addition timepoints). We are also performing a deeper structural analysis of the aggregates formed in presence and absence of the inhibitory effect (as also discussed in response to earlier reviewer comments). We prefer to publish those deeper mechanistic studies separately.

One additional comment: even in absence of inhibitors, the aggregation mechanism of HttEx1 is known to be complex and multifaceted, involving multiple different types of non-fibrillar species, and notable roles for both flanking domains (10.1038/nsmb.1570; 10.1073/pnas.1320626110; 10.7554/eLife.18065). For instance, there is convincing evidence that there are parallel aggregation pathways with a differing role of the N-terminal flanking domains (10.1021/bi3000929). This is complicating factor when aiming for a detailed mechanistic understanding, in particular when also considering possible interactions with these flanking domains (see also comments below).

The authors present evidence that compound treatment potentially changes the structure of the N- and/or C-terminal domains that are exposed on the surface of HttEx1 fibrils (the “fuzzy coat”). However, it remains unclear whether the compound binds directly to the exposed N17 or the PRD domain. This could be addressed using N- or C-terminally truncated HttEx1 fragments that both form amyloid structures in cell-free assays (see e.g., Mariscal et al., 2022 Nat Commun).

We thank the reviewer for this interesting question. As noted above, the flanking domains play important roles in the aggregation process. The presence or absence of N17 leads to qualitatively different aggregation processes, with dramatically different kinetics (see e.g. also our 2020 paper in JMB: 10.1016/j.jmb.2020.06.021). Moreover, the morphology of aggregates similarly changes. As such, extrapolating data observed with truncated versions of HttEx1 to what happens with HttEx1 itself, is not trivial (requiring careful control experiments, including a range of polypeptide constructs, structural studies, as well as numerous aggregation kinetics measurements).

Instead, to investigate the question of the reviewer we turned to solid-state NMR experiments that probe changes in the structure/dynamics of exposed flanking regions, which have the benefit that we can do so without need for removing parts of our protein. To do so, we prepared pre-formed ¹³C,¹⁵N-labeled HttEx1 fibrils (formed in absence of curcumin) and then treated these with curcumin after they were fully formed. These data are now included in Supplementary Figure 9 in the SI and discussed at the end of the Results section (page 13). The new figure compares the NMR spectra obtained on the original fibrils and after periods of curcumin

incubation, along with corresponding TEM data. The latter showed no large change in morphology due to curcumin treatment (i.e., no disaggregation or gross restructuring seems to be happening). Also the NMR spectra show no evidence of changes in the fibril core (polyQ). A notable feature in the curcumin-inhibited fibrils in the NMR spectra was a change in dynamics of the packed PRD domains on the fibril surface (Figure 4e). In the case of post-treatment with curcumin, we do not observe the same effect on the PRDs. Moreover, we see no detectable change in the dynamics or structure of the Htt^{NT} segment. Interestingly, we do see a small effect on the flexible C-terminal tail of HttEx1, but this effect is very small. We show these data in the Supplementary Figure 9 and discuss the findings on page 13 of the Results section (and briefly in the Discussion; page 18).

Also, dot blot assays should be performed with epitope-specific anti-HttEx1 antibodies to assess whether curcumin binding to HttEx1 amyloid fibrils alters their surface. Finally, compound binding studies with HttEx1 peptide arrays could be performed to better map the binding of curcumin to the “fuzzy coat”.

We thank the reviewer for the suggestion. In response to this comment, we tried studying curcumin binding of our fibrils with a dot blot assay similar to prior work of ours (Lin et al 2017 Nature Comm; 10.1038/ncomms15462). Unfortunately, the strong fluorescence of the curcumin interfered with our normal way of analyzing the dot blots, using a secondary antibody (Goat anti-Mouse IgG/IgM (H+L) that is an Alexa Flour 488 conjugate. The dot blot is shown below, for the information of the reviewer. We refer to the abovementioned NMR analysis as an alternative or complementary experiment to probe this question (and which did not show a large change in the exposed flexible C-terminus that is forming the fuzzy coat of the HttEx1 aggregates). We appreciate the mentioned peptide array studies as an interesting suggestion for future work, but they are beyond the scope of our current study.

The authors investigated the toxicity of compound treated and untreated HttEx1 fibrils using MTT assays. They state that HttEx1 fibril treatment influences cell viability. However, these initial observations need to be substantiated. MTT assays measure the metabolic activity of intracellular enzymes; MTT reduction may indicate cellular dysfunction and not necessarily reduced viability. Experiments with compound-treated and untreated HttEx1 fibrils should be performed in additional well-established cell-based assays, e.g., with the highly sensitive CellTiter-Glo and/or the CytoTox-Glo assays (commercially available).

The reviewer is right in emphasizing that the MTT assay measures the metabolic activity of the cells and, therefore, is not a direct measurement of cell viability. In our paper, we did not completely rely on the MTT assays for studying the toxicity of

HttEx1 fibrils. We also studied the effect of HttEx1 fibrils (formed with and without curcumin) on LUHMES cells using the Calcein AM staining experiment, in order to evaluate neuronal network integrity. This method is well established to evaluate cellular processes that lead to cell death, as neuronal network degeneration can be detected before the cell body damage (e.g. Ref. 10.1007/s00204-013-1072-y; 10.1016/j.biopha.2024.116163; 10.1016/j.freeradbiomed.2023.07.034). We concluded that the curcumin-fibrils caused less damage to the neuronal network. To clarify in the paper that we used two complementary methods, and to clarify the meaning of the different assays, we made changes on pages 9-10.

To further demonstrate that the assays chosen reflected the viability of the cells we analyzed nuclear morphology when LUHMES cells were exposed to 5 μ M HttEx1 fibrils for 24h (reference experiments only, without curcumin treatment). Fragmented nuclei are indicative of cells undergoing cellular death. Our data shows that LUHMES cells exposed to HttEx1 fibrils undergo cell death, and suggest that the loss in neuronal network integrity and metabolic activity are a consequence of cytotoxicity. These data are shown below, for the information of the reviewer.

(a) Representative pictures of differentiated LUHMES cells exposed for 24 hour to 5 μ M HttEx1 fibrils. Cells were fixed with 4% PFA, permeabilized with Triton-X-100 0.1%, and stained with DAPI (0.5 μ g/mL). An integer nuclear morphology was considered a viable cell, whereas fragmented nuclear morphology was considered undergoing cell death. White arrows highlight examples of viable nuclei, and yellow arrows highlight examples of dying nuclei. (b) Quantification of viable nuclei relative to the total number of nuclei.

Minor concerns:

Line 196: The unit "hours" differs within the manuscript between hours, h, hrs. Please unify.

We have re-written it to hours.

Line 205: A significant decrease is not shown for the condition 5 μ M/48 h.

We apologize for this error. We corrected the statement in line 205 to, "The highest concentration tested (25 μ M) led to a significant decrease in cell viability after 24 hour treatment (0.75 \pm 0.07; p<0.05), while all concentrations tested displayed a significant decrease in cell viability after 72 hour treatment." Main manuscript, page 9

Line 563: Spelling of Hoechst.

The spelling is corrected in the methods of the main manuscript, page 25.

Line 573: Check sentence.

This sentence in the methods (Flow cytometry) of the main manuscript, was in deed incomplete. This has been fixed on page 26 of the revised text.

Figure 2: (a) vs (b) > Why did you test different sub-stoichiometric ratios and why is the x-axis changed? Where are the data points? What is shown (a fitted curve)?

We changed this and other similar figures showing ThT fluorescence data to show the datapoints, rather than just a line-graph as was in the original. For completeness, we also included in Supplementary Figure 2 versions of each such figure that show the associated error bars.

Regarding the experimental conditions: these data come from experiments done in separate batches, and are mostly there to show the analogous effect of curcumin rather than enable a close comparison between the different Q-length proteins. Indeed, we regret that the conditions were not quite the same. We used fewer conditions with Q44-HttEx1 due to experimental challenges (see below), and therefore decided the higher ratio to fall between the values used for Q32-HttEx1.

A final comment, in our hands, these HttEx1 proteins are a bit challenging to handle due to their strong aggregation propensity, especially when the Q-stretch is 44 residues or longer. This requires preparations of fresh batches of protein for each experiment (as we cannot store the purified fusion protein in-between experiments) and expedited (and at times pragmatic) workflows. Given these challenges, we decided not to redo these experiments for a detailed Q32- vs Q44-HttEx1 comparison for this manuscript, which we consider not essential for the main findings of this paper.

Figure 2: (a) vs (b) > Why did you test different protein concentrations? (c-f) > 61 μM ? Why is the curcumin concentration in μM not ratio? Do black and white scale bars both represent 50 nm? How many fibrils did you measure? (m) > Red dashed line is missing.

In checking this comment, we discovered a mistakenly reported concentration in the original figure caption, which made the concentration difference seem larger than it really is (67 vs 61 μM). As noted above, these data were not designed to permit a direct comparison between the Q32- and Q44-HttEx1, but rather to illustrate the qualitatively similar effect of curcumin.

We modified the figure to show individual datapoints (as requested above) and also updated the markings in the figure to match the formatting used in Fig. 2a (i.e., showing the molar ratio as requested by the reviewer).

The black scale bars represent 100nm and white scale bar represents 50nm. We added this to the revised caption and it is also shown in the figure itself.

For the width analysis nearly 100 fibrils were measured. The fibril count is marked on the Y-axis of the figure panel.

Panel m – because this compares just two conditions, we did not think it was necessary to put the kind of line as we used to connect panels g-j.

Figure 4: Check labeling of subfigures.

The labeling has been corrected for figure 4.

Supplementary Figure 1: (d) > Understanding would be easier with same color code. Where are the data points?

Like the other ThT curves mentioned above, we now replaced the figure with versions in which the data points are visible (and not just a line plot). We also updated the color coding as suggested.

Extended Methods: ThT assay > Which bar graph in Figure 2b?

An originally existing bar graph in Figure 2b was previously removed. We have now deleted the erroneous comment from the Extended methods.

In conclusion, we want to thank the reviewers for their careful and detailed analysis of our paper and the constructive comments and feedback. The reviewer comments inspired us to clarify a number of points of the paper and to add new experimental data that provide additional mechanistic insights. We hope that the revised manuscript is significant improved, and now acceptable for publication.

Response to reviewer comments

Below we provide a point-by-point response to the reviewer comments, which we reproduce verbatim.

Reviewer #4

Jain et al.

Inhibitor-based modulation of huntingtin aggregation mechanisms reduces fibril toxicity

The authors have thoroughly addressed most of my concerns. One important point still needs clarification. The revised manuscript states (page 10, first paragraph): “Taken together, we observe that the mature HttEx1 fibrils exhibit toxic effects on both neuronal cell types, and that this toxicity is modulated significantly by curcumin inhibitors, resulting in less toxic fibril species.” This statement, however, is not in good agreement with the results shown in Fig. 3d. The quantification of neuronal network integrity (LUHMES cells) indicates that addition of Curcumin-treated fibrils does not significantly improve the neurite area (Fig. 3d). This needs to be clearly stated in the manuscript.

We appreciate the careful review by the reviewer. We revisited the data and our descriptions, and concluded that we agreed with the points made by this reviewer. We decided to adjust our interpretation of the cellular assays (also based on new data and other comments from the reviewer; see below). Our data clearly show that the protein fibrils cause increased stress for the neuronal cells that we have studied, resulting in measurable toxicity, which we analyze through multiple assays. Fibrils grown in presence of curcumin cause a significantly reduced stress burden on the cells, but we agree that the effects on measurements of ‘toxicity’ are less clear. (NB. In real patients, we may argue that long-term neuronal stress burden would or could translate into the type of slow neuronal degradation that characterizes HD and similar neurodegenerative disorders.)

In the revised text we make the above distinction much more clearly throughout the text, and decided to also adjust the paper title. To clarify the specific point that the fibrils formed in the presence of curcumin did not significantly prevent the loss of neuronal network integrity, also that part of the text was modified.

“The deterioration in network activity is evident by the reduction of the neurite area observed in treated human differentiated dopaminergic LUHMES neurons compared to non-treated controls. The group exposed to fibrils formed upon curcumin inhibition did not show a significant improvement in the neurite area. Nevertheless, there was a higher number of cell bodies positively stained with Calcein AM, which could reflect an improved viability, when compared to the group treated with fibrils formed in the absence of curcumin. Taken together, we observe that the mature HttEx1 fibrils exhibit toxic effects on both neuronal cell types, and that the cellular stress induced by the fibrils is mitigated by the effect of curcumin on fibril formation.”

Also, it still remains unclear whether the measured changes of MTT reduction in HT22 cells are an indication of reduced metabolic activity or reduced cell viability. In the figure legend (Fig. 3), the authors state that a cell viability assay was performed, while in the main text they state that MTT assays measure metabolic activity (page 9, line 197). It may well be that cell viability gets decreased when HT22 cells are treated with amyloidogenic HttEx1 aggregates.

Again, we appreciate the reviewer's comments, which highlight an important point. We agree that MTT reduction reflects cellular metabolic activity and may not always directly correlate with cell viability. First, we adjusted the caption of Figure 3 to remove references to viability assays (instead using the term MTT metabolic activity).

To be conclusive, this needs to be confirmed with an independent toxicity assay. As the manuscript's title states that inhibitor-based modulation of huntingtin aggregation reduces fibril toxicity, showing a convincing effect of Curcumin on fibril toxicity is prerequisite for the publication of this study.

To address this comment, and based on the comments from the reviewer in the prior round, we complemented the MTT assays with CytoTox-Glo assays. This assay was specifically proposed by the reviewer during the prior review round. This assay measures the release of cytosolic proteases due to loss of membrane integrity, which more directly indicates cell death, providing a robust assessment of cytotoxicity. Our data demonstrate that HttEx1 fibrils induce cell cytotoxicity in a dose-dependent manner (Supplementary Figure 3a). The addition of fibrils generated in the presence of curcumin presented a subtle mitigation of the cytotoxic effect (Supplementary Figure 3b and c). Combined, our results suggest that curcumin presence during fibril formation alters the cellular response to the HttEx1 fibrils in HT22 cells, which is most notable in the MTT assays, with potential beneficial effects on toxicity that are more subtle. To reflect these results in the title, we have changed it to focus more on the mitigating effect on cellular stress. We hope that these additional data, extensive clarifications and rephrasings, and changes in the title (and abstract) fully address the questions from this reviewer.

Reviewer #5

The authors describe work that shows that nucleation and growth of misfolded HttEx1 peptide fragments, into aggregates of fibers, is modulated by the polyphenol curcumin. In particular, fiber structure characterization by ssNMR, TEM and SAXS support the conclusion that HttEx1 aggregation with curcumin present results in the decrease in fiber width in a curcumin concentration dependent manner. The TEM study with micrographs and analysis displayed in figure 2 (for both Q44-HttEx1 and Q32-HttEx1 fibers) consistently shows a shift in fiber width distribution towards smaller values in the presence of curcumin with average width ranging from $\approx 10-14$ nm to $\approx 2-4$ nm.

The SAXS data appears consistent with the TEM data. However, I strongly recommend that the authors expand the discussion of the SAXS results and, in particular, in the context of three features as I detail below:

Indeed, our SAXS results are consistent with the TEM findings and provide complementary information. We are indebted to the reviewer for the valuable suggestions. We have

considered carefully the requests of the reviewer about expanding the discussion related to the SAXS results and we have added text to explain each structural feature of the SAXS curves in a more detailed way as explained below.

The SAXS data ($I(q)$ versus q) of HttEx1 aggregates in the absence and presence of curcumin (figure 2n) is displayed on a log-log plot. Three distinct regions can be identified.

At page 8, we have explicitly referred to the three regions mentioned by the reviewer, also stating the q -values for each range:

“Generally, three different regions can be detected: the low q region ($q < 0.2 \text{ nm}^{-1}$); the intermediate q region ($0.12 \text{ nm}^{-1} < q < 0.3 \text{ nm}^{-1}$); and the high q region ($q > 0.3 \text{ nm}^{-1}$), the so-called Porod-region.”

First, at low- q , the SAXS intensity exhibits power-law behavior (although over less than a decade) with $I(q) \approx 1/q$ indicative of a fractal of dimension 1 and consistent with fiber scattering. The main caution is that the power-law behavior is over a relatively narrow q range less than 1 decade (and even a shorter range for the fibers in the presence of curcumin). The authors should point out that the evidence of fiber structure up to much larger length scales is coming entirely from the TEM images. (the SAXS data at very low q is consistent with a linear fiber structure for length scales between $\approx \text{constant}/q_{\min}$ to $\approx \text{constant}/q^$, where q^* is the beginning of the so-called Porod regime where the slope decreases abruptly from -1 to $\approx -3-4$. The constant is between 1 and 2×3.14 depending on the model used.)*

We agree with the reviewer observation and we have accordingly added the following test at page 8 to explain this issue:

“In the low q region, the SAXS curves showed a power law decay of $q^{-\alpha}$. In the case of linear rod-like objects, an exponent $\alpha = 1$ is observed and the intensity follows the power law decay in the range $1/R < q < 2\pi/L$, where R and L are the radius and length of the rods. As shown in Figure 2n, the observed power law decay of the SAXS curves is consistent with the presence of linear fibrillar structures for length scales between $\approx 1/q^ = 1/(0.12 \text{ nm}^{-1}) \approx 8 \text{ nm}$ and $\approx 2\pi/q_{\min} = 1/(0.05 \text{ nm}^{-1}) \approx 125 \text{ nm}$, where q_{\min} is the minimum accessible q -value and q^* is the scattering vector value where the intensity slope in the log-log plot changes from the q^{-1} trend.”*

Second, the SAXS data clearly shows a correlation peak somewhere around $q \approx 0.3 \text{ nm}^{-1}$ (the lack of tick marks on the q -axis log plot makes it hard to see exactly where the peak is in q -space). The analysis of the scattering (in the supplement) via a model that incorporates both the form factor of rods and the structure factor measuring fiber-fiber correlations seems reasonable. The analysis gives an accurate measurement of the distance between fibers. However, the authors should include a brief discussion on the fact that the very large width of the correlation peak implies that there may only be near-neighbor correlations between fibers (this could either be coming from bundles with very few fibers or from bundles with many fibers but with defects such as kinks/breaks along the fiber length). The TEM images are a better method of determining the width of the bundles.

Following the reviewer’s suggestion, we have added the following text at page 8 to explain the broad nature of the correlation peak:

“Such a correlation peak is due to the existence of an average inter-fibrillar distance ($d_{f-f} = 2\pi/q_{max}$) as a consequence of the lateral aggregation (i.e., bundle formation) of the fibrils. The broad nature of this correlation peak implies that the correlation between fibrils is only short-ranged, and could either be coming from bundles formed by very few fibrils or from bundles formed by many fibrils but including a large amount of defects such as kinks or breaks along the fibril length.”

Third, the authors should point out that the SAXS intensity versus q exhibits a clear high q regime after the correlation peak (i.e. small length scales of order the width of the fiber and smaller) showing an abrupt decrease in slope from -1 to somewhere between -3 or -4 (i.e. the transitions from the linear fiber to the Porod regime). The rod form factor incorporated in the SAXS analysis accounts for this rapidly changing slope regime and gives an accurate measurement of the radius of the rods without curcumin and with curcumin (figure 2o).

Following the reviewer’s suggestion, we have added the following text at page 9 to comment about the high q region:

“In the high q region, after the correlation peak and for length scales of the order of the fibrils width and smaller, the SAXS intensity is dominated by the scattering contribution from the fibrils cross-section and the intensity decay changes from -1 to -3.2. The cross-sectional radius distribution and the lateral packing of the fibrils without and with curcumin can be accurately calculated by fitting the experimental intensity using the scattering function for long rod-like objects (see SI for details).”

Finally, another important issue which is straightforward to take care of: I suggest that the authors add tick marks (along the both the x and y axes) so that any knowledgeable SAXS person will be able to immediately deduce the slope of the scattering data, characteristic peak position related to inter-fiber distance, and an idea of the correlation length from the width of the peak (even before getting the parameters from a detailed modeling process involving the form factor and structure factor).

We recognize the lack of tick marks and we have now modified the panel in Figure 2n to include the minor marks so that readers can easily judge the structural features commented in our manuscript.

In conclusion, we want to thank the reviewers for their constructive comments and feedback, which inspired us to revisit our data, interpretations, terminology, and formulations in some detail. We hope that the revised manuscript is now acceptable for publication.

Response to reviewer comments

Below we provide a point-by-point response to the reviewer comments, which we reproduce verbatim.

Reviewer #4

Jain et al.

Inhibitor-based modulation of huntingtin aggregation mechanisms mitigates fibril-induced cellular stress

The authors have now comprehensively addressed all my concerns. The paper over time has improved significantly and is now ready for publication in Nature Communications. This study provides important mechanistic insights into the HttEx1 aggregation process as well as indicates that targeting of HttEx1 aggregation with the small molecule curcumin improves important molecular processes in cells.

We are glad to hear that our comprehensive edits have satisfied this reviewer.

Minor point:

The authors make the following statement in the text: "...there was a higher number of cell bodies positively stained with Calcein AM, which could reflect an improved viability, when compared to the group treated with fibrils formed in the absence of curcumin." However, it would be nice to show the results from the Calcein AM stainings in the Suppl. Figures to support this statement.

The reviewer asked for the inclusion of quantitative data on the Calcein AM staining assay. We have now added this information in Supplementary Figure 4, panel b.

Reviewer #5 (Remarks to the Author)

The authors have addressed all of my concerns. I recommend acceptance of their paper.